# Faster Query-Key Learning Sharpens Attention in Self-Attention Models

Rahul Vashisht [1]   Harish G. Ramaswamy [1] [2]

## Abstract

A standard self-attention layer consists of two interacting circuits: the query-key circuit that governs attention allocation, and the output-value circuit that maps attended representations to predictions. Collapsed and factorized parameterizations of the query-key and output-value circuits lead to qualitatively different attention patterns. In particular, some parameterizations give sharper attention to task-relevant tokens, at a similar training loss. We analyze how the parameterizations of these circuits shape the parameter trajectories in single-layer self-attention models trained on next-token prediction. Through gradient-flow analysis, we show that factorization induces implicit rescaling of the two circuits' learning rates. We derive closed-form dynamics showing that output-value and query-key parameters move along a line, with relative speeds determined by their learning rates. Faster query-key learning relative to output-value learning thus produces sharper attention, as the model compensates for slower output-value learning by increasing attention mass on relevant tokens. Experiments show that differences in the relative learning rates of the two circuits govern attention concentration. This improves attention interpretability proxies while maintaining comparable predictive performance.

## 1. Introduction

Transformer models (Vaswani et al., 2017) are now standard across language (Brown et al., 2020), vision (Dosovitskiy et al., 2020), and speech (Latif et al., 2023). Their success is driven by self-attention, which produces contextual representations and is typically trained using next-token prediction. Empirical studies have shown that trained models

with similar predictive performance can exhibit a wide range of attention patterns (Jain & Wallace, 2019; Wiegreffe & Pinter, 2019; Serrano & Smith, 2019). Prior work has also studied formal settings in which attention weights may or may not align with attention-based explanations (Pandey et al., 2023). However, the role of architectural and parameterization choices in the emergence of such attention patterns remains unclear.

Existing theoretical works have established that transformers are highly expressive, including universality and Turing completeness under suitable parameter settings (Yun et al., 2019; Bhattamishra et al., 2020a;b; Dehghani et al., 2018; Pérez et al., 2021). These results, however, rely on idealized parameter constructions and therefore do not explain how attention structure arises under standard gradient-based optimization. As a result, expressivity alone provides limited insight into how attention mechanisms and prediction layers co-evolve during training.

In this work, we analyze the training dynamics of a single-layer transformer trained for next-token prediction (Radford et al., 2018). The model decomposes naturally into two interacting linear circuits (Elhage et al., 2021; Olsson et al., 2022; Bietti et al., 2023):

- the **query-key circuit** ($W_K W_Q^\top$), which determines the attention pattern, and

- the **output-value circuit** ($W_O W_V$), which maps attended representations to predictions.

We show that the relative optimization speeds of these two circuits play a central role in shaping attention structure. Varying the learning rates of the query-key and output-value parameters leads to qualitatively different attention patterns, even when predictive performance remains similar. In particular, faster learning of the query-key circuit leads to sharper attention on task-relevant tokens.

A key focus of this paper is the role of parameterization. We compare factorized and collapsed parameterizations of both the query-key and output-value circuits and show that factorization induces a state-dependent rescaling of gradient updates in the corresponding collapsed parameter space. As a result, factorized and collapsed models can follow different optimization trajectories, even when they achieve

[1]Department of CSE, IIT Madras, India [2]Department of DSAI, IIT Madras, India. Correspondence to: Rahul Vashisht <rahul@cse.iitm.ac.in>, Harish G. Ramaswamy <hariguru@cse.iitm.ac.in>.

*Proceedings of the 43rd International Conference on Machine Learning*, Seoul, South Korea. PMLR 306, 2026. Copyright 2026 by the author(s).

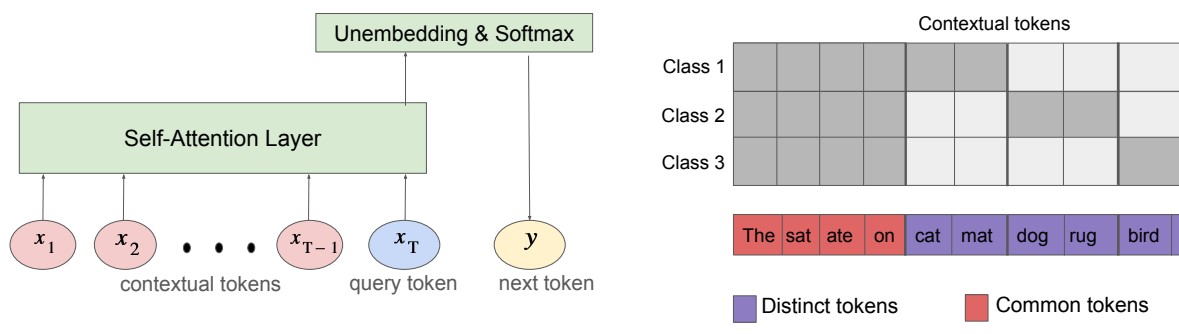

*(a)* Single Layer Self-Attention Model Illustration   *(b)* Synthetic Data Illustration

*Figure 1.* Overall Setting: (a) A sequence $\mathbf{X}$ with contextual tokens and query token $\{\boldsymbol{x}_1, \boldsymbol{x}_2, \ldots, \boldsymbol{x}_{T-1}, \boldsymbol{x}_T\}$ is fed into $1-$ layer transformer to predict the next token $y = \boldsymbol{x}_{T+1}$ (b) Synthetic data illustration, For each sequence class there exists a set of tokens specific to that class, called distinct tokens and a set of tokens common to all sequence classes called common tokens. For example 'the' 'sat' 'ate' and 'on' are the common tokens, and 'cat' 'mat' are tokens distinct to class 1

comparable losses.

We then characterize the training dynamics under collapsed parameterizations in a controlled synthetic setting. Under orthogonality and symmetry assumptions, we derive closed-form population gradient-flow dynamics for both circuits and show that their effective parameters evolve at different rates. In particular, the output-value scalar grows on the order of $\log(t)$, while the query-key scalar grows on the order of $\log^2(t)$, with constants controlled by the learning-rate ratio $\eta_{QK}/\eta_{OV}$. Because the query-key circuit affects attention through a saturating nonlinearity, this difference in growth provides a mechanistic explanation for how relative learning speeds shape attention sharpening during training.

We support the theory with experiments on synthetic data and real-world datasets with token-level relevance annotations, including HateXplain (Mathew et al., 2020), subject-verb agreement (Linzen et al., 2016), and SQuAD (Rajpurkar et al., 2016). Across settings, we observe that changes in relative optimization speed and parameterization alter attention structure in ways predicted by the theory, while predictive performance remains largely unchanged.

## 2. Related Works

Many recent works have studied the optimization dynamics of attention and self-attention-based models, particularly in simplified or single-layer settings (Tian et al., 2023; Deora et al., 2024a; Lu et al., 2021; Vashisht & Ramaswamy, 2023). Tian et al. (2023) analyze attention dynamics under the assumption that output–value parameters learn faster than query–key parameters. Deora et al. (2024a) study the optimization and generalization properties of multi-head self-attention models under realizability assumptions on the

data. Across these analyses, optimization dynamics are typically examined under fixed or implicit assumptions about the relative learning behavior of attention components. In contrast, we vary the relative learning rates of the query–key and output–value circuits and show that faster learning of query–key parameters sharpens attention. While prior work often focuses on single-sample or batch-size-one regimes for analytical tractability, we analyze full-batch training dynamics that more closely reflect modern practice. Moreover, we extend the analysis beyond the attention mechanism to the final prediction layer, providing a unified view of how interacting circuits shape the training trajectory.

A related line of work adopts a mechanistic perspective on transformer models. The transformer circuits framework of Elhage et al. (2021) provides a systematic decomposition of attention layers into interacting components, which has motivated subsequent analyses of how structure emerges in trained models, including clustering behavior in attention patterns (Geshkovski et al., 2023; Karagodin et al., 2024), memory-like mechanisms (Bietti et al., 2023), and the emergence of topic structure (Li et al., 2023). These works offer valuable insight into the functional roles of attention heads and learned representations after training. However, they primarily characterize attention structure post hoc and do not analyze how optimization dynamics shape the relative evolution of interacting circuits during training. Our work instead focuses on how the training trajectories of the query–key and output–value circuits jointly determine attention sharpening under gradient-based optimization.

## 3. Problem Setting

We study a single-layer self-attention model trained for next-token prediction. Let $\mathbf{X} = [\boldsymbol{x}_1, \ldots, \boldsymbol{x}_T]^\top \in \mathbb{R}^{T \times d}$ denote

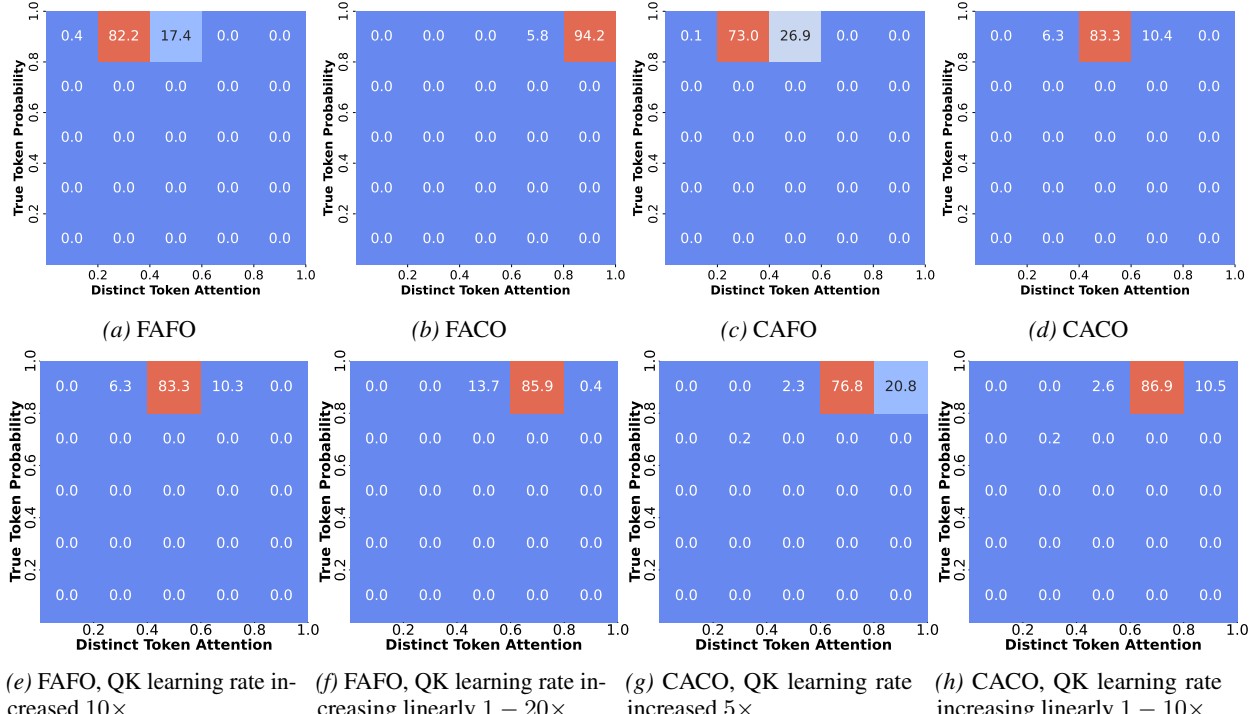

*(a) FAFO*     *(b) FACO*     *(c) CAFO*     *(d) CACO*

*(e) FAFO, QK learning rate increased 10×*    *(f) FAFO, QK learning rate increasing linearly 1 − 20×*    *(g) CACO, QK learning rate increased 5×*    *(h) CACO, QK learning rate increasing linearly 1 − 10×*

*Figure 2.* Distinct Token Attention-Prediction heat map of test data for different parameterizations under SGD with same learning rates (Top-Row) and Faster Learning for Query-Key (Bottom-Row). All models are trained until the negative log-likelihood is close to zero ($\leq 0.001$). Here, we train the model for 5 random seeds and report the average, refer to appendix for standard deviation in the heatmaps.

an input sequence, where $\boldsymbol{x}_T$ is the query token. The model parameters are $\boldsymbol{W}_Q, \boldsymbol{W}_K, \boldsymbol{W}_V \in \mathbb{R}^{d \times d}$ and $\boldsymbol{W}_O \in \mathbb{R}^{M \times d}$, where $M$ is the vocabulary size.

Given a sequence $\mathbf{X}$, the model output is

$$\boldsymbol{f}(\mathbf{X}) = \boldsymbol{W}_O \boldsymbol{W}_V \mathbf{X}^\top \mathbb{S}\big(\mathbf{X} \boldsymbol{W}_K \boldsymbol{W}_Q^\top \boldsymbol{x}_T\big), \qquad (1)$$

where $\mathbb{S}(\cdot)$ denotes the softmax operator. Figure 1a illustrates this computation for a single-layer self-attention model.

The model decomposes naturally into two linear circuits. The query-key circuit $\boldsymbol{W}_K \boldsymbol{W}_Q^\top$ determines the attention distribution over tokens, while the output-value circuit $\boldsymbol{W}_O \boldsymbol{W}_V$ maps the attended representation to prediction logits. We study four parameterizations that differ in whether these circuits are factorized or collapsed:

- **FAFO:** factorized attention and factorized output-value.

- **CAFO:** collapsed attention ($\boldsymbol{W}_{QK} = \boldsymbol{W}_K \boldsymbol{W}_Q^\top$) and factorized output-value.

- **FACO:** factorized attention and collapsed output-value ($\boldsymbol{W}_{OV} = \boldsymbol{W}_O \boldsymbol{W}_V$).

- **CACO:** collapsed attention and collapsed output-value.

Let $\boldsymbol{E} = [\mathbf{e}_1, \ldots, \mathbf{e}_M]^\top \in \mathbb{R}^{M \times d}$ denote the embedding matrix. Given a sequence $\mathbf{X}$ with next-token label $y \in [M]$, the training objective is to minimize the negative log-likelihood

$$\mathcal{L} = -\log \mathbb{S}_y(\boldsymbol{f}(\mathbf{X})). \qquad (2)$$

To analyze attention behavior in a controlled setting, we adopt a synthetic data construction based on Tian et al. (2023). Each sequence class is associated with a fixed query and next-token pair $(q, n)$, where the next token $n$ uniquely identifies the class. Context tokens preceding the query are sampled from a mixture of class-specific *distinct* tokens and class-agnostic *common* tokens, as illustrated in Figure 1b.

For example, consider three sequence classes:

- **Class 1**: "cat sat on the mat", "cat ate on the mat",

- **Class 2**: "dog sat on the rug", "dog ate on the rug",

- **Class 3**: "bird sat on the branch", "bird ate on the branch".

Here, tokens such as "sat", "ate", and "on" appear across all classes and are common, while tokens such as "cat", "mat", "dog", and "rug" are class-specific and distinct.

Formally, define $\Omega(l) = \{n : \mathbb{P}(l \mid n) > 0\}$. Tokens with $|\Omega(l)| = 1$ are distinct, while tokens with $|\Omega(l)| > 1$ are

common. Context tokens are sampled according to

$$\mathbb{P}(l \mid n) = \begin{cases} \gamma/C, & |\Omega(l)| = 1, \\ (1-\gamma)/D, & |\Omega(l)| > 1, \\ 0, & \text{otherwise,} \end{cases}$$

where $C$ and $D$ denote the number of distinct and common tokens, and $\gamma \in (0,1)$ controls the mixture weight. This construction separates informative and non-informative tokens and enables precise analysis of how attention and prediction evolve under different parameterizations and optimization speeds.

## 4. Self-Attention Analysis under Different Parameterizations

We identify an empirical phenomenon that motivates our theory. Different self-attention parameterizations exhibit qualitatively different attention patterns even when trained under identical optimization settings and achieving comparable predictive performance.

We consider a synthetic next-token prediction task with four sequence classes. Each sequence contains $m$ class-specific tokens, which are informative for prediction, $n$ common tokens have no class information, and a query token.

To assess alignment between attention and prediction, we introduce the Distinct Token Attention–Prediction (DTAP) heatmap. DTAP is a two-dimensional histogram in which the x-axis represents the fraction of attention assigned to class-specific tokens and the y-axis represents the predicted probability of the correct class. Each bin reports the fraction of examples that fall within the corresponding attention and confidence ranges. An ideal model concentrates most mass in the top-right corner, indicating both strong attention on informative tokens and high prediction confidence.

We compare four parameterizations: factorized attention–factorized output (FAFO), collapsed attention–collapsed output (CACO), collapsed attention–factorized output (CAFO), and factorized attention–collapsed output (FACO). Figure 2(a–d) shows that all parameterizations achieve high prediction confidence, yet their attention distributions differ substantially.

- Parameterizations with collapsed output–value structure (CACO and FACO) concentrate mass near the top-right region of the DTAP heatmap.

- The factorized output–value parameterization (FAFO) places substantial mass in the top-left region, corresponding to correct predictions made with weak attention on informative tokens.

- Increasing the learning rate of the query–key parameters shifts mass toward the top-right region across all

parameterizations.

As shown in Figure 2(e–h), accelerating query–key learning causes the attention patterns of FAFO and CACO to resemble those of FACO. In particular, attention mass shifts toward distinct tokens without degrading prediction accuracy. This behavior is observed under both fixed and gradually increasing query–key learning rates, indicating that relative optimization speed governs attention alignment beyond parameterization alone. These observations motivate the optimization analysis in Section 6.

The dataset for this task was generated as follows. We consider 4 sequence classes, with 4 query and corresponding next tokens. For each query token, we define a set of 10 distinct tokens. We also have a set of common tokens with cardinality 10. Note that the position (or index) of the distinct and common tokens can be arbitrary. We sample multiple such triples (context tokens, query token, next token), train a single-layer self-attention model on a subset of this dataset, and evaluate it on the remaining data. The vocabulary size is 50 tokens and sequence length is 64. We train the model with 200 batches with batch size 32 using stochastic gradient descent.

## 5. Isolating Output–Value Dynamics by Fixing Attention

The experiments in Section 4 show that both parameterization and relative learning rates influence attention allocation. To isolate the source of this effect, we analyze the learning dynamics of the output–value circuit while holding attention over distinct tokens fixed.

We consider a query–key circuit that assigns a total attention mass $\alpha \in [0, 1]$ to the distinct tokens, distributed uniformly among them, with the remaining mass assigned uniformly to the common tokens. Although fixing attention in this manner is not a realistic training regime, it serves as a diagnostic intervention that isolates the optimization behavior of the output–value circuit.

Under fixed $\alpha$, training the output–value parameters reduces to a multi-class classification problem with an effective input representation

$$\mathcal{L}(\alpha) = -\log\left(\text{softmax}_y\left(\boldsymbol{W}_O \boldsymbol{W}_V \widetilde{\boldsymbol{x}}\right)\right),$$

where

$$\widetilde{\boldsymbol{x}} = \frac{\alpha}{m} \sum_{j \in S_{\text{distinct}}} \boldsymbol{x}_j + \frac{1-\alpha}{n} \sum_{j \in S_{\text{common}}} \boldsymbol{x}_j.$$

As $\alpha$ increases, the effective input $\widetilde{\boldsymbol{x}}$ becomes more informative, simplifying the optimization problem faced by the

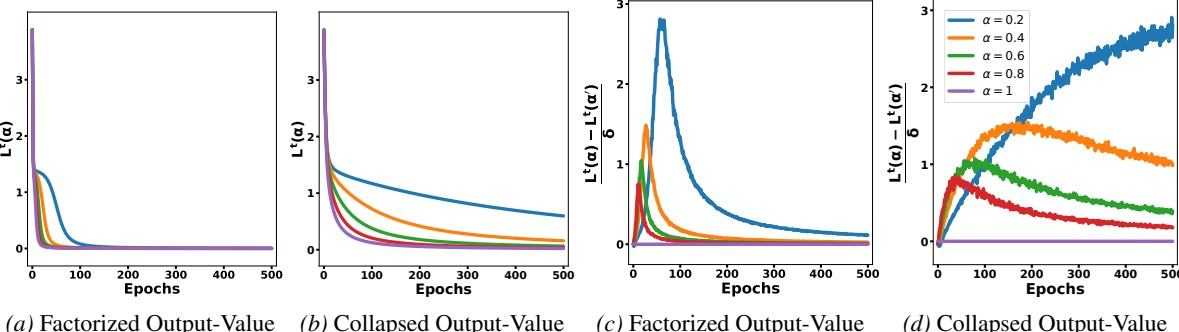

*(a)* Factorized Output-Value  *(b)* Collapsed Output-Value  *(c)* Factorized Output-Value  *(d)* Collapsed Output-Value

*Figure 3.* (a) and (b) Evolution of the training loss for fixed values of $\alpha$ under gradient descent. (c) and (d) show improvement incentive curves, measuring the change in loss resulting from a small increase $\delta = 0.005$ in $\alpha$.

output–value circuit. We define the fixed-attention loss curve as the training loss evaluated under gradient descent while holding the attention mass $\alpha$ fixed for distinct tokens. To quantify the incentive for attention sharpening, we measure the change in loss induced by a small increase in attention mass, $\delta = 0.005$, evaluated at a fixed training time.

Under the factorized output-value parameterization, the loss reaches near-zero at $\alpha = 0.2$ (blue curve in Figure 3a). The improvement incentive curve in Figure 3c shows a positive incentive for attention sharpening for up to 100 epochs, allowing $\alpha$ to increase to 0.4. However, for $\alpha \geq 0.4$ (orange curve), the incentive rapidly diminishes. As a consequence, attention tends to stabilize at values such as $\alpha = 0.4$.

In contrast, under the collapsed output-value parameterization, the loss does not decrease to zero as quickly for $\alpha \geq 0.2$ and $\alpha \leq 0.8$ (blue, orange, green, and red curves in Figure 3b). As a result, the incentive for increasing $\alpha$ remains high over a wider range, allowing attention to continue sharpening.

Taken together, the experiments in Sections 4 and 5 reveal an empirical pattern. Architectural parameterization shapes the relative learning dynamics of the query–key and output–value circuits, which in turn determines whether attention continues to sharpen or stabilizes early. These effects cannot be explained by predictive performance alone and persist across learning-rate interventions.

In the next section, we develop a theoretical framework that explains the observations by analyzing how different parameterizations induce distinct optimization trajectories under gradient flow.

# 6. Optimization Trajectories under Different Parameterization

Theoretical analyses of self-attention often simplify optimization dynamics (Deora et al., 2024b; Li et al., 2024; Bao et al., 2024) by working with collapsed parameter matrices, rather than explicitly modeling the factorized query–key and output–value parameters (Elhage et al., 2021; Olsson et al., 2022). This simplification enables tractable analysis, but it is not a priori clear whether it faithfully reflects the training dynamics of factorized models. Empirically, as shown in Section 4, factorized and collapsed parameterizations can achieve comparable predictive performance while exhibiting qualitatively different attention behaviors.

In a single-layer self-attention model, each circuit consists of a composition of linear transformations with no intervening nonlinearity. Thus, the optimization dynamics of the query-key and output-value circuits fall within the class of homogeneous linear models for which conservation laws under gradient flow hold (Arora et al., 2018; Du et al., 2018). Building on results from deep linear network theory, we show that factorized training induces a well-defined dynamics on the collapsed parameters, differing from collapsed training only through a state-dependent preconditioning operator.

Let $z = W_{OV} X^\top a$ and $\widetilde{z} = W_O W_V X^\top a$. We define the loss functions $\mathcal{L}^1(W_{OV}) = -\log \mathbb{S}_y(z)$, $\mathcal{L}^2(W_O, W_V) = -\log \mathbb{S}_y(\widetilde{z})$, where the attention weights are given by $a = \mathbb{S}(X W_K W_Q^\top x_T)$.

We now relate the gradient flow dynamics of the factorized model to those of the collapsed parameterization. Let $W_O(t)$ and $W_V(t)$ denote the gradient flow trajectories under $\mathcal{L}^2$, and define the effective matrix $\widetilde{W}_{OV}(t) = W_O(t) W_V(t)$.

The proposition 1 is a direct application of the chain rule.

**Proposition 1.** *Let $C = AB$, where $A$, $B$, and $C$ are matrices of shape $M \times d$, $d \times d$ and $M \times d$.*

*Then the gradients satisfy*

$$\nabla_{W_O} \mathcal{L}^2 \big|_{(A,B)} = \nabla_{W_{OV}} \mathcal{L}^1 \big|_C B^\top,$$
$$\nabla_{W_V} \mathcal{L}^2 \big|_{(A,B)} = A^\top \nabla_{W_{OV}} \mathcal{L}^1 \big|_C. \qquad (3)$$

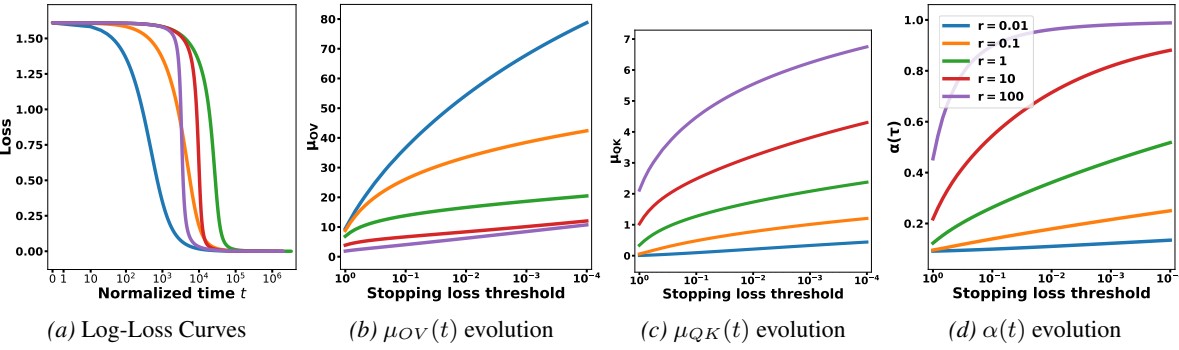

*(a)* Log-Loss Curves     *(b)* $\mu_{OV}(t)$ evolution     *(c)* $\mu_{QK}(t)$ evolution     *(d)* $\alpha(t)$ evolution

*Figure 4.* Population gradient-flow simulations for the orthogonal synthetic setting of Theorem 1 with $m = 5$, $n = 50$, $b = 50$, $M = 5$, varying the ratio $r = \eta_{QK}/\eta_{OV}$. (a) Training loss vs. normalized time, (b) $\mu_{OV}(t)$ at matched stopping loss, (d) $\mu_{QK}(t)$ at matched stopping loss, and attention mass $\alpha(t)$ evaluated at matched stopping losses.

The gradient identities in Proposition 1 do not by themselves imply a closed evolution for the effective matrix $\widetilde{W}_{OV} = W_O W_V$, since the dynamics of $W_O$ and $W_V$ may depend on their relative scaling. We therefore next characterize a structural condition under which factorized gradient flow admits a well-defined induced dynamics on the effective parameters.

**Lemma 1.** *Let $W_O(t)$ and $W_V(t)$ denote the gradient flow trajectories with loss $\mathcal{L}^2$. Let $W_O(0)^\top W_O(0) = W_V(0) W_V(0)^\top$, Then $W_O(t)^\top W_O(t) = W_V(t) W_V(t)^\top$*

Lemma 1 ensures that the evolution of the effective matrix does not depend on arbitrary rescalings of the factorized parameters. Under this condition, the factorized gradient flow induces a closed evolution for the effective matrix that depends only on the collapsed loss.

**Lemma 2.** *Let $W_O(t)$ and $W_V(t)$ follow gradient flow on $\mathcal{L}^2$, and define the effective matrix $\widetilde{W}_{OV}(t) = W_O(t) W_V(t)$. Assume $W_O(0)^\top W_O(0) = W_V(0) W_V(0)^\top$. Then $\widetilde{W}_{OV}(t)$ evolves according to the preconditioned gradient flow for $\mathcal{L}^1$*

$$\frac{d}{dt} \operatorname{vec}(\widetilde{W}_{OV}(t)) = -\Omega(\widetilde{W}_{OV}(t)) \operatorname{vec}\left(\nabla \mathcal{L}^1(\widetilde{W}_{OV}(t))\right),$$

*where $\Omega(C) = (C^\top C)^{1/2} \oplus (C C^\top)^{1/2}$,*

$\Omega(C)$ *a mapping from $\mathbb{R}^{M \times d} \to \mathbb{R}^{Md \times Md}$ is a degree 1-homogeneous map.* $\oplus$ *denotes the Kronecker sum.*

Lemma 2 shows that, under the balancedness condition, the evolution of the effective matrix $\widetilde{W}_{OV}(t)$ induced by factorized training is fully determined by the collapsed loss $\mathcal{L}^1$, up to the application of a state-dependent preconditioning operator. In contrast to collapsed training, which follows standard gradient flow on $\mathcal{L}^1$, factorized training follows a preconditioned gradient flow in the same parameter space. $\Omega$ being a homogeneous map implies that using factorized

parameterization is closely approximated by collapsed factorization with a learning rate that increases over time.

The same argument applies to the query-key factorization $\widetilde{W}_{QK} = W_K W_Q^\top$. Its gradient flow admits an analogous preconditioned representation of the collapsed query-key dynamics. We provide the full derivation in Appendix D. Together, these results justify analyzing optimization trajectories in the collapsed parameter space for both circuits, provided the induced preconditioning is taken into account.

### 6.1. Synthetic Orthogonal Data Setting

This section isolates the mechanism by which relative learning rates between the query-key and output-value circuits shape attention during training. We introduce a simplified synthetic setting that enables a transparent, closed-form characterization of how attention sharpening emerges under gradient flow. To enable such a characterization, we make the following assumptions on the data distribution.

**Assumption 1.** *For each class $y \in [M]$, the class-specific token distribution is supported on $b$ distinct tokens, $D_y = \frac{1}{b} \sum_{\tau=1}^{b} \delta(\mathbf{s}_y^\tau)$, where for each $\tau \in [b]$, the vectors $\mathbf{s}_1^\tau, \mathbf{s}_2^\tau, \ldots, \mathbf{s}_M^\tau \in \mathbb{R}^d$ are mutually orthogonal.*

**Assumption 2.** *The background token distribution $D_0$ has zero mean.*

**Assumption 3.** *The support of $D_0$ is orthogonal to all class-specific token vectors $\{\mathbf{s}_y^\tau\}_{y \in [M], \tau \in [b]}$.*

Together, these assumptions eliminate cross-token interactions, reducing training to a low-dimensional system in which the coupling between attention learning and output learning can be analyzed explicitly.

Each training sequence consists of $m$ class-specific token positions (sampled with replacement from the $b$ tokens of the corresponding class), $n$ background token positions, and a final query token, giving total length $T = m + n + 1$. The attention vector is $\boldsymbol{\alpha} = \operatorname{softmax}(X W_{QK} x_T)$, and

we denote by $\alpha(t) := \sum_{j \in \mathcal{D}} \alpha_j$ the total attention mass on class-specific token positions (relevant token for a class).

The following theorem shows that, under Assumptions 1-3, training dynamics collapse to two coupled scalar trajectories governing output-value learning and attention sharpening.

**Theorem 1.** *Under the orthogonal full-data setting and initialization $\boldsymbol{W}_{OV}(0) = \boldsymbol{W}_{QK}(0) = \boldsymbol{0}$, the output-value and query-key parameters evolve under population gradient flow with learning rates $\eta_{OV}$ and $\eta_{QK}$ as*

$$\boldsymbol{W}_{OV}^{(k,:)}(t) = \mu_{OV}(t) \left[ \sum_{\tau=1}^{b} s_k^{\tau} - \frac{1}{M} \sum_{y=1}^{M} \sum_{\tau=1}^{b} s_y^{\tau} \right],$$

$$\boldsymbol{W}_{QK}^{(k,:)}(t) = \mu_{QK}(t) \sum_{y=1}^{M} \left( \sum_{\tau=1}^{b} s_y^{\tau} \right) \left( \sum_{\tau'=1}^{b} s_y^{\tau',k} \right),$$

*where the scalar coefficients satisfy*

$$\frac{d\mu_{OV}(t)}{dt} = \frac{\eta_{OV}\, \alpha(t)}{b\big( \exp(\alpha(t)\mu_{OV}(t)) + M - 1 \big)},$$

$$\frac{d\mu_{QK}(t)}{dt} = \frac{\eta_{QK}\, (M-1)\big(\alpha(t) - \alpha(t)^2\big)\mu_{OV}(t)}{Mb^2\big( \exp(\alpha(t)\mu_{OV}(t)) + M - 1 \big)},$$

*with*

$$\alpha(t) = \frac{m\, \exp(\mu_{QK}(t))}{m\, \exp(\mu_{QK}(t)) + n}.$$

In particular, both parameter matrices evolve along fixed data-dependent directions, with all learning dynamics captured by the scalar coefficients $\mu_{OV}(t)$ and $\mu_{QK}(t)$.

**Lemma 3** (Relative growth rates (pre-saturation regime))**.** *Under the population gradient-flow dynamics of Theorem 1, assume that the attention mass satisfies $\alpha(t) \leq 1 - \delta$ for some $\delta > 0$ over a time interval $[0, T]$. Then for all $t \in [0, T]$,*

$$\mu_{OV}(t) = \Theta\big( \ln(1+t) \big), \quad \mu_{QK}(t) = \Theta\left( \frac{\eta_{QK}}{\eta_{OV}} (\ln(1+t))^2 \right).$$

Lemma 3 formalizes the core mechanism: the output-value scale grows logarithmically in time, while the growth of the query-key scale that governs attention concentration is controlled by the learning-rate ratio $r = \eta_{QK}/\eta_{OV}$ through a log-squared dependence. When output learning is slowed relative to attention learning ($\eta_{QK} > \eta_{OV}$), optimization compensates by increasing the magnitude of the query-key parameters, thereby concentrating attention mass on class-relevant tokens.

Figure 4a shows that the training loss converges to zero for all values of the ratio $r = \eta_{QK}/\eta_{OV}$, indicating that final predictive accuracy alone does not distinguish between learning regimes. Figure 4b shows the evolution of $\mu_{OV}$ evaluated at matched stopping losses varying values of $r$. Figures 4c and d similarly show the evolution of $\mu_{QK}$ and the attention mass $\alpha$ at matched stopping loss thresholds.

To understand the differing attention behaviors, we examine the intermediate dynamics in Fig. 4b–c. Figure 4b shows the evolution of $\mu_{OV}(t)$, which serves as a proxy for logit growth and directly governs loss decay, while Fig. 4c shows the evolution of $\mu_{QK}(t)$, which acts as a proxy for growth of attention on relevant tokens.

For all values of $r$, $\mu_{OV}(t)$ increases sufficiently to drive the loss to zero, explaining the similar convergence behavior observed in Fig. 4a. In contrast, $\mu_{QK}(t)$ depends strongly on $r$: for large ratios (e.g., $r = 100$), $\mu_{QK}(t)$ grows rapidly, leading to attention mass on relevant tokens close to 1, whereas for small ratios (e.g., $r = 0.01$), $\mu_{QK}(t)$ remains small, resulting in attention close to uniform despite loss becoming zero. This decoupling between logit growth and attention sharpening shows that selective attention emerges from the relative optimization dynamics of the $QK$ and $OV$ rather than from improved prediction.

## 7. Experiments

The goal of our experiments is to test qualitative predictions of the theory developed in Section 6. We do not aim to optimize task performance or explore architectural design choices. Instead, we evaluate whether changing the relative learning rates of the query-key and output-value circuits produces the attention behavior predicted by the theory.

The theory makes two core predictions. First, increasing the ratio $r = \dfrac{\eta_{QK}}{\eta_{OV}}$ should lead to sharper attention. Second, this attention sharpening should occur while predictive performance remains comparable. All experiments in this section are designed as controlled interventions on this learning rate ratio.

We compare a baseline transformer to a variant trained in a higher query-key learning-rate regime. In this variant, we increase the learning rate of the query-key parameters while keeping all other settings fixed. This directly increases the ratio $r$. This intervention corresponds directly to the mechanism analyzed in Lemma 3.

We evaluate three datasets with token-level relevance annotations: SQuAD, HateXplain, and subject-verb agreement (SVA). For SQuAD, we use a 6-layer GPT-style model. For HateXplain and SVA, we use 1-layer models to match the theoretical setting. Additional multi-layer results are provided in the Appendix D.5. Here, predictive performance

*Table 1.* Train and test performance together with attention-based interpretability metrics on real-world datasets. ↑ indicates higher is better, while ↓ indicates lower is better. All results are reported as average over 5 runs. We report 95% confidence intervals over five random seeds in Table 4.

| Dataset | Setting | Train Performance ↑ | Test Performance ↑ | AC ↑ | ACMC ↑ | MRTA (Rollout) ↑ | Suff. ↓ | Compre. ↑ |
|---|---|---|---|---|---|---|---|---|
| SQuAD QA | Baseline | 70.30 | 56.11 | 12.65 | 12.6 | 0.06 | 0.69 | 0.38 |
| | Faster QK | 70.81 | 56.86 | 42.95 | 41.48 | 0.1 | 0.59 | 0.53 |
| SVA | Baseline | 95.02 | 92.434 | 41.67 | 40.4 | 0.1 | 0.24 | 0.24 |
| | Faster QK | 95.938 | 93.412 | 76.85 | 74.47 | 0.19 | 0.15 | 0.30 |
| HateXplain | Baseline | 88.75 | 55.74 | 0.00 | 0.00 | 0.027 | 0.41 | 0.40 |
| | Faster QK | 89.72 | 56.90 | 43.5 | 30.29 | 0.31 | 0.32 | 0.48 |

refers to the standard task metric for each dataset: F1 score for SQuAD and classification accuracy for HateXplain and SVA.

All models are trained with AdamW. We first tune a single base learning rate using validation performance. This rate is shared across all parameters in the baseline model. In the increased $r$ regime, we multiply the query-key learning rate by a fixed factor while keeping the output-value learning rate unchanged. We emphasize that adaptive optimization alone does not induce this behavior, as adaptive optimizers treat all parameter groups equally. We tune the query–key multiplier using validation performance to ensure stable training and comparable predictive accuracy across settings. Importantly, this tuning is not used to optimize attention metrics. For concreteness, we use base learning rates of $5 \times 10^{-5}$, $5 \times 10^{-5}$, and $1 \times 10^{-5}$ for SQuAD, HateXplain, and SVA respectively, and increase the query-key learning rate by fixed multipliers of $20\times$, $30\times$, and $100\times$ while keeping the output-value learning rate unchanged. Hyperparameter details are in Appendix A.2.

We also report attention-based metrics that directly test the theoretical predictions. We measure mean relevant token attention (MRTA) and use attention rollout (Abnar & Zuidema, 2020) to aggregate the scores across layers.

We report sufficiency and comprehensiveness scores as faithfulness metrics (DeYoung et al., 2020). Sufficiency measures if important tokens are enough to retain the original prediction (lower is better), while comprehensiveness measures the change in probability of the predicted class after removing important tokens (higher is better).

Finally, we use DTAP-based summary metrics. Attention Confidence (AC) measures the percentage of instances in which a large fraction of attention mass is assigned to task-relevant tokens. Attention Confidence & Model Confidence (ACMC) measures the percentage of confidently correct instances that also exhibit high attention concentration. These metrics are designed to capture the increase in attention mass $\alpha(t)$ predicted by the theory. We discuss these metrics in detail in Appendix D.6.

Table 1 summarizes the results across all datasets. Predictive

performance remains stable across tasks. Increasing $r$ does not degrade test performance, and in some cases yields small improvements. This confirms that attention sharpening is not driven by improved prediction alone.

At the same time, attention becomes substantially sharper. Both AC and ACMC increase by large margins across all datasets. Sufficiency decreases and comprehensiveness increases in all settings, indicating more faithful attention. These effects are consistent across datasets and persist in deeper models. They match the qualitative behavior predicted by the optimization dynamics in Section 6.

Overall, the results support the theoretical mechanism proposed in this work. When output-value learning is slower, optimization increases the query-key scale. This concentrates attention through the softmax nonlinearity. The model compensates for slower output learning by attending more strongly to informative tokens. Importantly, this effect does not require larger output weights or improved predictive performance. It arises from relative optimization speeds alone.

## 8. Conclusion

In this work, we study how optimization dynamics shape attention structure in self-attention models. By analyzing the interaction between the query-key and output-value circuits, we showed that attention sharpening is controlled by the relative learning speeds of these components rather than by predictive performance alone. In particular, faster learning of the query-key circuit leads to sharper attention, even when performance is comparable. Our theoretical analysis provides a mechanistic explanation for this behavior, which is supported by experiments on synthetic data and real-world benchmarks. These results highlight the importance of parameterization and learning-rate choices. These factors are often treated as implementation details but play a central role in shaping attention behavior. Our findings suggest that attention interpretability can be systematically influenced through optimization dynamics, without requiring architectural changes.

# 9. Limitations

The theoretical analysis in the paper focuses on a single-layer attention-only transformer. Although our empirical results suggest that the core findings extend to standard multi-layer transformers, additional components such as layer normalization and feedforward networks may introduce dynamics that require further theoretical study. Our conclusions are also tied to the simplifying assumptions used in the theory and experiments. Sharper attention or improved alignment with annotation-based metrics does not necessarily imply a fully trustworthy explanation of the model's reasoning, especially in high-stakes settings. The learning-rate intervention should be viewed as one concrete probe of relative QK-OV optimization dynamics rather than as the only possible causal mechanism. Other circuit-specific interventions, such as initialization imbalance, weight decay, or gradient normalization, are left for future work. We do not claim that sharper attention directly improves downstream task performance. Our results show that attention structure can be controlled, with improvements on attention-based interpretability proxies.

# Impact Statement

This work studies how optimization dynamics influence attention patterns in self-attention models. A potential positive impact is improved understanding of internal model behavior for interpretability and downstream performance. However, sharper attention or improved alignment with annotation-based metrics does not necessarily imply faithful explanation, and should not be used on its own in high-stakes settings.

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

# A. Supplementary Material

## A.1. Code Reproducibility

Code for reproducing the experiments is available at https://github.com/vashishtrahul/Faster-Query-Key-Learning-Sharpens-Attention-in-Self-Attention-Models.

## A.2. Dataset Details

### A.2.1. SYNTHETIC DATA SECTION 4

The dataset for the task in Section 4 is generated as follows. We consider 4 sequence classes, with 4 query and corresponding next tokens. For each query token, we define a set of 10 distinct tokens. We also have a set of common tokens with cardinality 10. Note that the position (or index) of the distinct and common tokens can be arbitrary. We sample multiple such triples (context tokens, query token, next token), train a single-layer self-attention model on a subset of this dataset, and evaluate it on the remaining data. The vocabulary size is 50 tokens and sequence length is 64. We train the model with 200 batches with batch size 32 using stochastic gradient descent.

### A.2.2. REAL-WORLD DATASET SECTION 7

Table 2 lists the three datasets used in the experiments.

*Table 2.* Dataset statistics: number of instances and rationales for each split.

| Dataset | Train Inst | Validation Inst. | Test Inst. | Number of Rationales |
|---|---|---|---|---|
| Hatexplain | 15383 | 1922 | 1924 | 9132 (train), 1141 (test/validation) |
| Subject-Verb Agreement | 100K | 20K | 20K | 100K (Train), 20K (Test/Validation) |
| SQuAD QA | 40K | 10K | - | - |

Table 3 gives model architecture details for each dataset.

*Table 3.* Model architecture configurations used in experiments.

| Dataset | Variant | $d_{\text{model}}$ | $n_{\text{heads}}$ | $d_{\text{head}}$ | $d_{\text{mlp}}$ / $d_{\text{ff}}$ | $n_{\text{layers}}$ | $n_{\text{ctx}}$ |
|---|---|---|---|---|---|---|---|
| HateXplain | 1-layer (baseline) | 64 | 4 | 64 | 256 | 1 | 256 |
| SVA | 1-layer | 64 | 4 | 64 | 256 | 1 | 256 |
| HateXplain | 2-layer (baseline) | 32 | 4 | 8 | 128 | 2 | 96 |
| HateXplain | 4-layer | 64 | 4 | 64 | 256 | 4 | 256 |
| SQuAD | GPTAnswerGenerator | 300 | 6 | 50 | 1200 | 6 | 256 |

**HateXplain Data**: We describe the validation and test results for the HateXplain dataset here. For the preprocessing of text data, we follow (Mathew et al., 2020) and clean the data using the same method. For hyperparameter tuning, we first consider the same-learning-rate setting and then use the same initialization for the faster-learning setting, except for the query-key parameters, for which we use a learning rate of $30\times$ the base rate of $5e{-}5$.

**Hyperparameter Tuning**: We choose the base learning rate by hyperparameter tuning on validation data for learning rates in range $[0.01, 0.000001]$. Once we choose the base learning rate, we increase the learning rate of query-key parameters by $5\times, 10\times, 20\times, 30\times$, and $100\times$. In the main paper, we observe that increasing the learning rate of the query-key parameters improves interpretability in terms of heatmaps for all the cases. We report the best performing hyperparameters in the paper. For this data, we also experiment with $2-$layer and $4-$layer models.

**Subject Verb Agreement Data**: We further test our hypothesis on the Subject-Verb Agreement (SVA) dataset (binary-classification) (Linzen et al., 2016), which evaluates the model's ability to capture long-range dependencies between subjects and verbs. The dataset contains sentences designed to check grammatical number agreement (e.g., whether a singular

subject like "the cat" pairs with a singular verb like "runs," or a plural subject like "the cats" pairs with "run"). This task provides a controlled setting to examine whether faster learning for query-key parameters, which improved interpretability for HateXplain, generalizes to core syntactic tasks. We use the same single-layer attention-only model. We train the model with SGD and batch size 256, and consider two settings: a same learning-rate setting and a faster-query-key-learning setting. The validation and test results for SVA are described here. For the preprocessing of text data, we follow (Linzen et al., 2016) and clean the data using the same method. For hyperparameter tuning, we first consider the same-learning-rate setting and then use the same weights for the faster-learning setting, except for the query-key parameters, for which we use a learning rate of $100\times$ the base rate of $1\text{e}-5$.

**Hyperparameter Tuning**: We choose the base learning rate by hyperparameter tuning on validation data for learning rates in range $[0.01, 0.000001]$. Once we choose the base learning rate, we increase the learning rate of query-key parameters by $5\times$, $10\times$, $20\times$, $30\times$, and $100\times$. In the main paper, we observe that increasing the learning rate of the query-key parameters improves interpretability in terms of heatmaps for all the cases. We report the best performing hyperparameters in the paper.

**SQuAD QA Data**: We also run large-scale experiments on the subset of SQuAD question-answering dataset. Here, we finetune a 6-layer GPT model to generate answers based on given contexts and questions. We use the answer span information only for evaluation of attention-based interpretability. We train the model under two settings: (1) same learning rate for all parameters and (2) faster learning for query-key parameters ($20\times$ the base rate of $5\text{e}-5$). We also show the results on proxies for attention-based interpretability after aggregating attention scores using the rollout method (Abnar & Zuidema, 2020).

**Hyperparameter Tuning**: We choose the base learning rate by hyperparameter tuning on validation data for learning rates in range $[0.01, 0.00001]$. Once we choose the base learning rate, we increase the learning rate of query-key parameters by $5\times$, $10\times$, $20\times$, $30\times$, and $100\times$. In the main paper, we observe that increasing the learning rate of the query-key parameters improves interpretability proxies for interpretability in terms of heatmaps for all the cases. We report the best performing hyperparameters in the paper.

All DTAP heatmaps averaged over 5 runs are provided in Appendix D.5.

**Standard Deviation Table for Table 1**

*Table 4.* We report 95% confidence intervals, shown as $\pm$ values, for each metric in Table 1.

| Dataset | Setting | Train Performance ↑ | Test Performance ↑ | AC ↑ | ACMC ↑ | MRTA (Rollout) ↑ | Suff. ↓ | Compr. ↑ |
|---|---|---|---|---|---|---|---|---|
| SQuAD QA | Baseline | 0.71 | 0.63 | 2.34 | 2.41 | 0.003 | 0.012 | 0.03 |
| | Faster QK | 0.61 | 0.40 | 7.29 | 6.68 | 0.01 | 0.07 | 0.03 |
| SVA | Baseline | 0.84 | 0.03 | 0.29 | 0.31 | 0.0001 | 0.17 | 0.14 |
| | Faster QK | 0.60 | 0.15 | 8.45 | 8.29 | 0.03 | 0.06 | 0.01 |
| HateXplain | Baseline | 0.87 | 1.04 | 0.00 | 0.00 | 0.00 | 0.06 | 0.06 |
| | Faster QK | 0.01 | 2.28 | 1.18 | 2.07 | 0.01 | 0.05 | 0.03 |

# B. Theorem 1 Proof

**Theorem 1** [Parameter Trajectories]    Under the orthogonal full-data setting, consider sequences of length $T$ with $m$ distinct tokens and $n$ common tokens. Suppose that $\boldsymbol{W}_{OV}(0) = 0$ and $\boldsymbol{W}_{QK}(0) = 0$, and let $\eta_{OV}$ and $\eta_{QK}$ be the learning rates of $\boldsymbol{W}_{OV}$ and $\boldsymbol{W}_{QK}$, respectively. Then the parameters evolve under population gradient flow as follows:

$$\boldsymbol{W}_{OV}^{(k,:)}(t) = \mu_{OV}(t)\left[\sum_{\tau=1}^{b} s_k^\tau - \frac{1}{M}\sum_{y=1}^{M}\sum_{\tau=1}^{b} s_y^\tau\right],$$

$$\boldsymbol{W}_{QK}^{(k,:)}(t) = \mu_{QK}(t)\left[\sum_{y=1}^{M}\sum_{\tau=1}^{b} s_y^\tau \sum_{\tau'=1}^{b} s_y^{\tau',k}\right],$$

where the scalars $\mu_{OV}(t)$ and $\mu_{QK}(t)$ vary as follows:

$$\frac{d\mu_{OV}(t)}{dt} = \frac{\eta_{OV}\alpha(t)}{b * (\exp(\alpha(t)\mu_{OV}(t)) + M - 1)},$$

$$\frac{d\mu_{QK}(t)}{dt} = \frac{\eta_{QK}(M-1)(\alpha(t) - \alpha^2(t))\mu_{OV}(t)}{b^2 M (\exp(\alpha(t)\mu_{OV}(t)) + M - 1)},$$

$$\alpha(t) = \frac{m \times \exp(\mu_{QK}(t))}{m * (\exp(\mu_{QK}(t))) + n}.$$

*Proof.* We consider the following form of the network output:

$$\boldsymbol{z} = \boldsymbol{W}_{OV}\, \widetilde{\boldsymbol{x}}, \qquad \widetilde{\boldsymbol{x}} = \boldsymbol{X}^\top \boldsymbol{\alpha}, \qquad \boldsymbol{\alpha} = \text{softmax}(\boldsymbol{X}\boldsymbol{W}_{QK}\boldsymbol{x}_T),$$

where $\boldsymbol{W}_{OV} \in \mathbb{R}^{M \times d}$, $\boldsymbol{W}_{QK} \in \mathbb{R}^{d \times d}$, $\boldsymbol{X} \in \mathbb{R}^{T \times d}$ stacks the sequence tokens, and $\boldsymbol{x}_T \in \mathbb{R}^d$ is the query token.

**Gradient of cross-entropy with softmax.**

Let $\boldsymbol{z} = \boldsymbol{W}_{OV}\boldsymbol{X}^\top\boldsymbol{\alpha}$ and $\boldsymbol{p} = \text{softmax}(\boldsymbol{z})$. For the cross-entropy loss $\mathcal{L} = -\log \boldsymbol{p}_y$,

$$-\nabla_{\boldsymbol{W}_{OV}^{(k,:)}}\mathcal{L} = \left(\mathbf{1}[y = k] - \boldsymbol{p}_k\right)\boldsymbol{X}^\top\boldsymbol{\alpha}.$$

Under population gradient flow (averaging over the data distribution),

$$-\nabla_{\boldsymbol{W}_{OV}^{(k,:)}}\mathcal{L} = \mathbb{E}\left[\left(\mathbf{1}[y = k] - \boldsymbol{p}_k\right)\sum_{j=1}^{T}\boldsymbol{\alpha}_j\boldsymbol{x}_j^\top\right]. \tag{4}$$

Note that $\boldsymbol{\alpha}$ depends on the sampled sequence. Using linearity of expectation,

$$\mathbb{E}\left[\sum_{j=1}^{T}\boldsymbol{\alpha}_j\boldsymbol{x}_j\right] = \sum_{j=1}^{T}\mathbb{E}[\boldsymbol{\alpha}_j\boldsymbol{x}_j].$$

Let $\mathcal{D}$ denote the $m$ distinct-token positions and $\mathcal{C}$ denote the $n$ common-token positions. Define the scalar

$$\alpha(t) := \sum_{j \in \mathcal{D}}\boldsymbol{\alpha}_j,$$

i.e., the total attention mass on distinct-token positions. By symmetry, all distinct positions have the same expected weight, and all common positions have the same expected weight:

$$\mathbb{E}[\boldsymbol{\alpha}_j] = \begin{cases} \alpha(t)/m, & j \in \mathcal{D}, \\ (1 - \alpha(t))/n, & j \in \mathcal{C}. \end{cases}$$

Under the full-data symmetry assumptions for distinct-token sampling,

$$\mathbb{E}\left[\sum_{j \in \mathcal{D}}\boldsymbol{x}_j \,\middle|\, y\right] = \frac{m}{b}\sum_{\tau=1}^{b}s_y^\tau.$$

Moreover, since the common-token distribution has zero mean, $\mathbb{E}[\nu] = 0$, we have $\mathbb{E}\left[\sum_{j \in \mathcal{C}}\boldsymbol{\alpha}_j\boldsymbol{x}_j\right] = 0$.

Therefore,

$$\mathbb{E}[\boldsymbol{X}^\top \boldsymbol{\alpha} \mid y] = \mathbb{E}\left[\sum_{j \in \mathcal{D}} \boldsymbol{\alpha}_j \boldsymbol{x}_j \,\middle|\, y\right] = \frac{\alpha(t)}{m} \mathbb{E}\left[\sum_{j \in \mathcal{D}} \boldsymbol{x}_j \,\middle|\, y\right] = \frac{\alpha(t)}{b} \sum_{\tau=1}^{b} s_y^\tau. \tag{5}$$

Substituting (5) into (4) yields

$$-\nabla_{\boldsymbol{W}_{OV}^{(k,:)}} \mathcal{L} = \frac{1}{M} \sum_{y=1}^{M} \left(\mathbf{1}[y = k] - \boldsymbol{p}_k\right) \frac{\alpha(t)}{b} \sum_{\tau=1}^{b} s_y^\tau. \tag{6}$$

Now consider the ansatz

$$\boldsymbol{W}_{OV}^{(k,:)}(t) = \mu_{OV}(t) \left[\sum_{\tau=1}^{b} s_k^\tau - \frac{1}{M} \sum_{y'=1}^{M} \sum_{\tau=1}^{b} s_{y'}^\tau\right].$$

Let

$$\widetilde{\boldsymbol{s}}_k := \sum_{\tau=1}^{b} s_k^\tau - \frac{1}{M} \sum_{y'=1}^{M} \sum_{\tau=1}^{b} s_{y'}^\tau.$$

Using orthogonality of the vectors $\{s_y^\tau\}_{y,\tau}$,

$$\widetilde{\boldsymbol{s}}_k^\top \sum_{\tau=1}^{b} s_y^\tau = \left(\sum_{\tau=1}^{b} s_k^\tau\right)^\top \left(\sum_{\tau=1}^{b} s_y^\tau\right) - \frac{1}{M} \left(\sum_{y'=1}^{M} \sum_{\tau=1}^{b} s_{y'}^\tau\right)^\top \left(\sum_{\tau=1}^{b} s_y^\tau\right)$$

$$= b \, \mathbf{1}[k = y] - \frac{b}{M} = b\left(\mathbf{1}[k = y] - \frac{1}{M}\right). \tag{7}$$

Hence, for a sequence with label $y$,

$$\boldsymbol{W}_{OV} \, \mathbb{E}[\boldsymbol{X}^\top \boldsymbol{\alpha} \mid y] = \mu_{OV}(t) \frac{\alpha(t)}{b} \begin{bmatrix} \widetilde{\boldsymbol{s}}_1^\top \sum_{\tau=1}^{b} s_y^\tau \\ \vdots \\ \widetilde{\boldsymbol{s}}_M^\top \sum_{\tau=1}^{b} s_y^\tau \end{bmatrix} = \mu_{OV}(t)\alpha(t)\left(e_y - \frac{1}{M}\mathbf{1}\right). \tag{8}$$

Using shift-invariance of softmax, this implies

$$\boldsymbol{p} = \text{softmax}(\mu_{OV}(t)\alpha(t) \, e_y),$$

and hence

$$\boldsymbol{p}_{k=y} = \frac{\exp(\alpha(t)\mu_{OV}(t))}{\exp(\alpha(t)\mu_{OV}(t)) + M - 1}, \qquad \boldsymbol{p}_{k \neq y} = \frac{1}{\exp(\alpha(t)\mu_{OV}(t)) + M - 1}. \tag{9}$$

Using (9) into (6), we have,

$$-\nabla_{\boldsymbol{W}_{OV}^{(k,:)}} \mathcal{L} = \frac{1}{M} \sum_{y=1}^{M} \left(\mathbf{1}[y = k] - \boldsymbol{p}_k\right) \frac{\alpha(t)}{b} \sum_{\tau=1}^{b} s_y^\tau. \tag{10}$$

$$-\nabla_{\boldsymbol{W}_{OV}^{(k,:)}} \mathcal{L} = \frac{\alpha(t)}{b * M} \left[\left[1 - \boldsymbol{p}_{k=y}\right] \sum_{\tau=1}^{b} s_k^\tau + \sum_{y=1, y \neq k}^{M} -\boldsymbol{p}_{k \neq y} \sum_{\tau=1}^{b} s_y^\tau\right] \tag{11}$$

$$-\nabla_{\boldsymbol{W}_{OV}^{(k,:)}}\mathcal{L} = \frac{\alpha(t)}{b*M}\left[\left[1 - \frac{\exp(\alpha(t)\mu_{OV}(t))}{\exp(\alpha(t)\mu_{OV}(t)) + M - 1}\right]\sum_{\tau=1}^{b} s_k^{\tau} \right.$$
$$\left. - \sum_{y=1,y\neq k}^{M}\frac{1}{\exp(\alpha(t)\mu_{OV}(t)) + M - 1}\sum_{\tau=1}^{b} s_y^{\tau}\right].$$

$$-\nabla_{\boldsymbol{W}_{OV}^{(k,:)}}\mathcal{L} = \frac{\alpha(t)}{b*M}\left[\left[\frac{M-1}{\exp(\alpha(t)\mu_{OV}(t)) + M - 1}\right]\sum_{\tau=1}^{b} s_k^{\tau} \right.$$
$$\left. - \sum_{y=1,y\neq k}^{M}\frac{1}{\exp(\alpha(t)\mu_{OV}(t)) + M - 1}\sum_{\tau=1}^{b} s_y^{\tau}\right].$$

$$-\nabla_{\boldsymbol{W}_{OV}^{(k,:)}}\mathcal{L} = \frac{\alpha(t)}{b*M(\exp(\alpha(t)\mu_{OV}(t)) + M - 1)}\left[[M-1]\sum_{\tau=1}^{b} s_k^{\tau} - \sum_{y=1,y\neq k}^{M}\sum_{\tau=1}^{b} s_y^{\tau}\right].$$

$$-\nabla_{\boldsymbol{W}_{OV}^{(k,:)}}\mathcal{L} = \frac{\alpha(t)}{b*M(\exp(\alpha(t)\mu_{OV}(t)) + M - 1)}\left[M\sum_{\tau=1}^{b} s_k^{\tau} - \sum_{y=1}^{M}\sum_{\tau=1}^{b} s_y^{\tau}\right].$$

$$-\nabla_{\boldsymbol{W}_{OV}^{(k,:)}}\mathcal{L} = \frac{\alpha(t)}{b(\exp(\alpha(t)\mu_{OV}(t)) + M - 1)}\left[\sum_{\tau=1}^{b} s_k^{\tau} - \frac{1}{M}\sum_{y=1}^{M}\sum_{\tau=1}^{b} s_y^{\tau}\right].$$

$$\boxed{-\nabla_{\boldsymbol{W}_{OV}^{(k,:)}}\mathcal{L} = \frac{\alpha(t)}{b\left(\exp(\alpha(t)\mu_{OV}(t)) + M - 1\right)}\,\widetilde{\boldsymbol{s}}_k}. \tag{12}$$

Therefore, under gradient flow,

$$\frac{d\boldsymbol{W}_{OV}^{(k,:)}(t)}{dt} = \eta_{OV}\frac{\alpha(t)}{b\left(\exp(\alpha(t)\mu_{OV}(t)) + M - 1\right)}\,\widetilde{\boldsymbol{s}}_k. \tag{13}$$

On the other hand, differentiating the ansatz $\boldsymbol{W}_{OV}^{(k,:)}(t) = \mu_{OV}(t)\widetilde{\boldsymbol{s}}_k$ with respect to time yields

$$\frac{d\boldsymbol{W}_{OV}^{(k,:)}(t)}{dt} = \frac{d\mu_{OV}(t)}{dt}\,\widetilde{\boldsymbol{s}}_k.$$

Since $\widetilde{\boldsymbol{s}}_k \neq 0$ and the same direction $\widetilde{\boldsymbol{s}}_k$ appears in (13), we obtain the scalar differential equation

$$\frac{d\mu_{OV}(t)}{dt} = \frac{\eta_{OV}\,\alpha(t)}{b\left(\exp(\alpha(t)\mu_{OV}(t)) + M - 1\right)}.$$

**Key-Query Parameter Trajectory Proof**

Recall $\widetilde{x} = X^{\top}\boldsymbol{\alpha}$ and $\boldsymbol{\alpha} = \mathrm{softmax}(\boldsymbol{X}\boldsymbol{W}_{QK}x_T)$. Let $\boldsymbol{z} = \boldsymbol{W}_{OV}\widetilde{x}$ and $\boldsymbol{p} = \mathrm{softmax}(\boldsymbol{z})$.

$$-\nabla_{\boldsymbol{W}_{QK}^{(k,:)}}\mathcal{L} = \frac{1}{M}\sum_{y=1}^{M}\boldsymbol{x}_T\sum_{j=1}^{T}\boldsymbol{\alpha}_j\,\boldsymbol{x}_{j,k}\,(\boldsymbol{x}_j - \widetilde{\boldsymbol{x}})^{\top}\left[\boldsymbol{W}_{OV}^{(y,:)} - \sum_{k'=1}^{M}\boldsymbol{p}_{k'}\,\boldsymbol{W}_{OV}^{(k',:)}\right].\tag{14}$$

Using (8), the logits satisfy

$$\boldsymbol{W}_{OV}(\boldsymbol{X}^{\top}\boldsymbol{\alpha}) = \mu_{OV}(t)\alpha(t)\left[\boldsymbol{e}_y - \frac{1}{M}\mathbf{1}\right],$$

and therefore

$$\boldsymbol{W}_{OV}^{(y,:)} - \sum_{k'=1}^{M}\boldsymbol{p}_{k'}\,\boldsymbol{W}_{OV}^{(k',:)} = \frac{M\mu_{OV}(t)}{\exp(\alpha(t)\mu_{OV}(t)) + M - 1}\left[\sum_{\tau=1}^{b}s_y^{\tau} - \frac{1}{M}\sum_{k'=1}^{M}\sum_{\tau'=1}^{b}s_{k'}^{\tau'}\right].\tag{15}$$

We now analyze the remaining factor in (14) under population expectation. Let $\mathcal{D}$ denote the multiset of distinct-token positions in the sequence ($|\mathcal{D}| = m$, counting multiplicities), and $\mathcal{C}$ denote the multiset of common-token positions ($|\mathcal{C}| = n$).

Under the trajectory ansatz for $W_{QK}$,

$$W_{QK}^{(k,:)}(t) = \mu_{QK}(t)\sum_{y=1}^{M}\left(\sum_{\tau=1}^{b}s_y^{\tau}\right)\left(\sum_{\tau'=1}^{b}s_y^{\tau',k}\right),$$

the attention scores $\boldsymbol{u}_j = \boldsymbol{x}_j^{\top}\boldsymbol{W}_{QK}\boldsymbol{x}_T$ take only two values: for distinct-token positions $j \in \mathcal{D}$, $\boldsymbol{u}_j = \mu_{QK}(t)$, while for common-token positions $j \in \mathcal{C}$, $\boldsymbol{u}_j = 0$. Consequently, the softmax attention vector $\boldsymbol{\alpha}$ is uniform within each group:

$$\boldsymbol{\alpha}_j = \begin{cases} \alpha(t)/m, & j \in \mathcal{D}, \\ (1 - \alpha(t))/n, & j \in \mathcal{C}, \end{cases} \qquad \alpha(t) = \sum_{j \in \mathcal{D}}\boldsymbol{\alpha}_j.$$

Writing $\boldsymbol{x}_j = s_y^{\tau}$ for distinct tokens and $\boldsymbol{x}_j = \nu^{\tau}$ for common tokens, the population expectation of the first factor in (14) becomes

$$\frac{1}{M}\sum_{y=1}^{M}\boldsymbol{x}_T\sum_{j=1}^{T}\alpha_j\,\boldsymbol{x}_{j,k}\,(\boldsymbol{x}_j - \widetilde{\boldsymbol{x}})^{\top} = \frac{1}{M}\sum_{y=1}^{M}\left(\frac{1}{b}\sum_{r=1}^{b}s_y^r\right)\left\{\frac{\alpha(t)}{m}\sum_{\tau=1}^{b}s_y^{\tau,k}\left(s_y^{\tau} - \frac{\alpha(t)}{m}\sum_{\tau'=1}^{b}s_y^{\tau'} - \frac{1-\alpha(t)}{n}\sum_{\tau''=1}^{n}\nu^{\tau''}\right)^{\top}\right.$$
$$\left. + \frac{1-\alpha(t)}{n}\sum_{\tau=1}^{n}\nu^{\tau,k}\left(\nu^{\tau} - \frac{\alpha(t)}{m}\sum_{\tau'=1}^{b}s_y^{\tau'} - \frac{1-\alpha(t)}{n}\sum_{\tau''=1}^{n}\nu^{\tau''}\right)^{\top}\right\}.\tag{16}$$

Substituting (15) and (16) into (14) yields

$$-\nabla_{\boldsymbol{W}_{QK}^{(k,:)}}\mathcal{L} = \frac{\mu_{OV}(t)}{\exp(\alpha(t)\mu_{OV}(t)) + M - 1}\left[\sum_{y=1}^{M}\left(\frac{1}{b}\sum_{r=1}^{b}s_y^r\right)\{\cdots\}\right]\left[\sum_{\tau=1}^{b}s_y^{\tau} - \frac{1}{M}\sum_{k'=1}^{M}\sum_{\tau'=1}^{b}s_{k'}^{\tau'}\right],\tag{17}$$

where $\{\cdots\}$ denotes the bracketed expression in (16).

Under the orthogonality assumptions and the zero-mean condition $\mathbb{E}[\nu^{\tau}] = 0$, all cross terms involving common tokens vanish under the population expectation. The expression therefore reduces to the distinct-token component:

$$-\nabla_{\boldsymbol{W}_{QK}^{(k,:)}}\mathcal{L} = \frac{\mu_{OV}(t)}{\exp(\alpha(t)\mu_{OV}(t)) + M - 1}\sum_{y=1}^{M}\left(\frac{1}{b}\sum_{r=1}^{b}s_y^r\right)\widetilde{C}_y\,V_y,\tag{18}$$

where

$$\widetilde{C}_y = \frac{\alpha(t)}{m} \sum_{\tau=1}^{b} s_y^{\tau,k} \left( s_y^{\tau} - \frac{\alpha(t)}{m} \sum_{\tau'=1}^{b} s_y^{\tau'} \right)^{\top}, \tag{19}$$

$$V_y = \left[ \sum_{\tau=1}^{b} s_y^{\tau} - \frac{1}{M} \sum_{k'=1}^{M} \sum_{\tau'=1}^{b} s_{k'}^{\tau'} \right]. \tag{20}$$

Using orthogonality of $\{s_y^{\tau}\}$, the product $\left( \frac{1}{b} \sum_{r=1}^{b} s_y^{r} \right) \widetilde{C}_y V_y$ simplifies to a rank-one term, yielding

$$\boxed{ -\nabla_{\boldsymbol{W}_{QK}^{(k,:)}} \mathcal{L} = \frac{\mu_{OV}(t)(M-1)\alpha(t)\big(1-\alpha(t)\big)}{Mb^2\big(\exp(\alpha(t)\mu_{OV}(t)) + M - 1\big)} \sum_{y=1}^{M} \left( \sum_{\tau=1}^{b} s_y^{\tau} \right) \left( \sum_{\tau'=1}^{b} s_y^{\tau',k} \right). } \tag{21}$$

In continuous time,

$$\frac{d\boldsymbol{W}_{QK}^{(k,:)}}{dt} = \eta_{QK} \frac{\mu_{OV}(t)(M-1)\alpha(t)\big(1-\alpha(t)\big)}{Mb^2\big(\exp(\alpha(t)\mu_{OV}(t)) + M - 1\big)} \sum_{y=1}^{M} \left( \sum_{\tau=1}^{b} s_y^{\tau} \right) \left( \sum_{\tau'=1}^{b} s_y^{\tau',k} \right), \tag{22}$$

and hence

$$\frac{d\mu_{QK}(t)}{dt} = \eta_{QK} \frac{\mu_{OV}(t)(M-1)\alpha(t)\big(1-\alpha(t)\big)}{Mb^2\big(\exp(\alpha(t)\mu_{OV}(t)) + M - 1\big)}. \tag{23}$$

Finally, since distinct-token positions have score $\mu_{QK}(t)$ and common-token positions have score $0$, the total attention mass on distinct-token positions is

$$\alpha(t) = \frac{m \exp(\mu_{QK}(t))}{m \exp(\mu_{QK}(t)) + n}.$$

$\square$

## C. Lemma 3 Proof

**Lemma 3** [Part I: $\mu_{OV}(t)$ Bounds]   Let $\mu_{OV}(t)$ evolve as

$$\dot{\mu}_{OV}(t) = \frac{\eta_{OV}\, \alpha(t)}{b\big(e^{\alpha(t)\mu_{OV}(t)} + M - 1\big)}, \qquad \mu_{OV}(0) = 0,$$

where $\alpha(t) = \frac{me^{\mu_{QK}(t)}}{me^{\mu_{QK}(t)} + n}$ and $\alpha_0 := \alpha(0) = \frac{m}{m+n}$. Then for all $t \geq 0$,

$$\log\Big(1 + \frac{\eta_{OV}\alpha_0}{Mb}\, t\Big) \ \leq\ \mu_{OV}(t) \ \leq\ \frac{1}{\alpha_0} \log\Big(1 + \frac{\eta_{OV}\alpha_0}{b}\, t\Big). \tag{24}$$

*Proof.* From $\dot{\mu}_{QK}(t) \geq 0$ and $\mu_{QK} = 0$, we have $\mu_{QK}(t) \geq 0$ and non-decreasing; since $\alpha()$ is increasing, $\alpha(t)$ is non-decreasing and $\alpha_0 \leq \alpha(t) \leq 1$

For any $x \geq 0$, $e^x \leq e^x + M - 1 \leq Me^x$, hence

$$\frac{1}{M} e^{-x} \leq \frac{1}{e^x + M - 1} \leq e^{-x}.$$

Apply this with $x = \alpha(t)\mu_{OV}(t)$ in the ODE to obtain

$$\frac{\eta_{OV}\alpha(t)}{Mb} e^{-\alpha(t)\mu_{OV}(t)} \ \leq\ \dot{\mu}_{OV}(t) \ \leq\ \frac{\eta_{OV}\alpha(t)}{b} e^{-\alpha(t)\mu_{OV}(t)}. \tag{25}$$

Using $\alpha(t) \leq 1$ and $e^{-\alpha(t)\mu} \leq e^{-\alpha_0 \mu}$ (since $\alpha(t) \geq \alpha_0$ and $\mu \geq 0$), the right inequality of (25) implies

$$\dot{\mu}_{OV}(t) \leq \frac{\eta_{OV}}{b} e^{-\alpha_0 \mu_{OV}(t)}.$$

Multiplying by $\alpha_0 e^{\alpha_0 \mu_{OV}(t)}$ gives

$$\alpha_0 e^{\alpha_0 \mu_{OV}(t)} \dot{\mu}_{OV}(t) \leq \frac{\eta_{OV}\alpha_0}{b},$$

i.e.,

$$\frac{d}{dt}\left(e^{\alpha_0 \mu_{OV}(t)}\right) \leq \frac{\eta_{OV}\alpha_0}{b}.$$

Integrating from 0 to $t$ and using $e^{\alpha_0 \mu_{OV}(0)} = 1$ yields

$$e^{\alpha_0 \mu_{OV}(t)} \leq 1 + \frac{\eta_{OV}\alpha_0}{b} t.$$

Applying $\log(\cdot)$ and dividing by $\alpha_0$ gives

$$\mu_{OV}(t) \leq \frac{1}{\alpha_0} \log\left(1 + \frac{\eta_{OV}\alpha_0}{b} t\right).$$

Using $\alpha(t) \geq \alpha_0$ and $e^{-\alpha(t)\mu} \geq e^{-\mu}$ (since $\alpha(t) \leq 1$ and $\mu \geq 0$), the left inequality of (25) implies

$$\dot{\mu}_{OV}(t) \geq \frac{\eta_{OV}\alpha_0}{Mb} e^{-\mu_{OV}(t)}.$$

Multiplying by $e^{\mu_{OV}(t)}$ gives

$$e^{\mu_{OV}(t)} \dot{\mu}_{OV}(t) \geq \frac{\eta_{OV}\alpha_0}{Mb},$$

i.e.,

$$\frac{d}{dt}\left(e^{\mu_{OV}(t)}\right) \geq \frac{\eta_{OV}\alpha_0}{Mb}.$$

Integrating from 0 to $t$ and using $e^{\mu_{OV}(0)} = 1$ yields

$$e^{\mu_{OV}(t)} \geq 1 + \frac{\eta_{OV}\alpha_0}{Mb} t.$$

Applying $\log(\cdot)$ gives

$$\mu_{OV}(t) \geq \log\left(1 + \frac{\eta_{OV}\alpha_0}{Mb} t\right).$$

Combining the two bounds proves (24). $\qquad\square$

**Lemma 3** [Part II: $\mu_{QK}(t)$ Bounds]  Let $\mu_{OV}(t), \mu_{QK}(t)$ satisfy the coupled ODEs

$$\dot{\mu}_{OV}(t) = \frac{\eta_{OV}\,\alpha(t)}{b\left(e^{\alpha(t)\mu_{OV}(t)} + M - 1\right)}, \qquad \dot{\mu}_{QK}(t) = \eta_{QK} \frac{(M-1)\alpha(t)(1-\alpha(t))\mu_{OV}(t)}{Mb^2\left(e^{\alpha(t)\mu_{OV}(t)} + M - 1\right)},$$

with $\mu_{OV}(0) = \mu_{QK}(0) = 0$ and $\alpha(t) = \frac{me^{\mu_{QK}(t)}}{me^{\mu_{QK}(t)}+n}$. Let $\alpha_0 = \frac{m}{m+n}$.

For all $t \geq 0$,

$$\mu_{QK}(t) \ \leq \ \frac{\eta_{QK}}{\eta_{OV}} \frac{(M-1)(1-\alpha_0)}{2Mb\,\alpha_0^2} \log^2\left(1 + \frac{\eta_{OV}\alpha_0}{b} t\right). \tag{26}$$

Fix $\delta \in (0,1)$ and assume

$$\alpha(t) \leq 1 - \delta \qquad \text{for all } t \in [0,T]. \tag{27}$$

Then for all $t \in [0,T]$,

$$\mu_{QK}(t) \ \geq \ \frac{\eta_{QK}}{\eta_{OV}} \frac{(M-1)\delta}{2Mb} \log^2\left(1 + \frac{\eta_{OV}\alpha_0}{Mb} t\right). \tag{28}$$

*Proof.* Divide the $\mu_{QK}$-ODE by the $\mu_{OV}$-ODE to cancel the common denominator:

$$\frac{d\mu_{QK}}{d\mu_{OV}} = \frac{\dot{\mu}_{QK}}{\dot{\mu}_{OV}} = \frac{\eta_{QK}}{\eta_{OV}} \frac{M-1}{Mb} (1 - \alpha(t)) \, \mu_{OV}(t). \tag{29}$$

Since $\alpha(t)$ is nondecreasing and $\alpha(0) = \alpha_0$, we have $\alpha(t) \geq \alpha_0$ and hence $1 - \alpha(t) \leq 1 - \alpha_0$. Using this in (29) yields

$$\frac{d\mu_{QK}}{d\mu_{OV}} \leq \frac{\eta_{QK}}{\eta_{OV}} \frac{M-1}{Mb} (1 - \alpha_0) \, \mu_{OV}.$$

Integrating over $\mu_{OV} \in [0, \mu_{OV}(t)]$ gives

$$\mu_{QK}(t) \leq \frac{\eta_{QK}}{\eta_{OV}} \frac{M-1}{Mb} (1 - \alpha_0) \int_0^{\mu_{OV}(t)} u \, du = \frac{\eta_{QK}}{\eta_{OV}} \frac{(M-1)(1-\alpha_0)}{2Mb} \mu_{OV}(t)^2.$$

Applying the upper bound from Lemma 3,

$$\mu_{OV}(t) \leq \frac{1}{\alpha_0} \log\left(1 + \frac{\eta_{OV}\alpha_0}{b} t\right),$$

we obtain

$$\mu_{QK}(t) \leq \frac{\eta_{QK}}{\eta_{OV}} \frac{(M-1)(1-\alpha_0)}{2Mb\,\alpha_0^2} \log^2\left(1 + \frac{\eta_{OV}\alpha_0}{b} t\right),$$

which is exactly (26).

Assume (27). Then $1 - \alpha(t) \geq \delta$ for all $t \in [0, T]$. Using this in (29) yields

$$\frac{d\mu_{QK}}{d\mu_{OV}} \geq \frac{\eta_{QK}}{\eta_{OV}} \frac{M-1}{Mb} \delta \, \mu_{OV}.$$

Integrating over $\mu_{OV} \in [0, \mu_{OV}(t)]$ gives

$$\mu_{QK}(t) \geq \frac{\eta_{QK}}{\eta_{OV}} \frac{M-1}{Mb} \delta \int_0^{\mu_{OV}(t)} u \, du = \frac{\eta_{QK}}{\eta_{OV}} \frac{(M-1)\delta}{2Mb} \mu_{OV}(t)^2.$$

Applying the lower bound of $\mu_{OV}$ from Lemma 3,

$$\mu_{OV}(t) \geq \log\left(1 + \frac{\eta_{OV}\alpha_0}{Mb} t\right),$$

we obtain

$$\mu_{QK}(t) \geq \frac{\eta_{QK}}{\eta_{OV}} \frac{(M-1)\delta}{2Mb} \log^2\left(1 + \frac{\eta_{OV}\alpha_0}{Mb} t\right),$$

which is exactly (28). $\qquad\square$

## D. Relation Between Collapsed and Factorized Parameterization

In this section, we provide proofs for Proposition 1, Lemma 1, and Lemma 2. we also state the corresponding proposition and lemmas for QK circuit in proposition 2, 4 and 5.

### D.1. Proposition 1 Proof

**Proposition 1** Consider the network output

$$\widehat{y} = \boldsymbol{W}_O \boldsymbol{W}_V \widetilde{x}, \qquad \boldsymbol{W}_O \in \mathbb{R}^{M \times d}, \ \boldsymbol{W}_V \in \mathbb{R}^{d \times d}, \ \widetilde{x} \in \mathbb{R}^d,$$

and define the collapsed output-value parameter $\boldsymbol{W}_{OV} := \boldsymbol{W}_O \boldsymbol{W}_V$. Let $\widetilde{y} = \mathbb{S}(\widehat{y})$ and let $\mathcal{L}$ be the cross-entropy loss

$$\mathcal{L} = -\log\left(\frac{\exp(\widehat{y}_c)}{\sum_k \exp(\widehat{y}_k)}\right).$$

Then,

$$\frac{\partial \mathcal{L}}{\partial \boldsymbol{W}_{OV}} = (\widetilde{y} - y)\,\widetilde{x}^\top. \tag{30}$$

By chain rule,

$$\frac{\partial \mathcal{L}}{\partial \boldsymbol{W}_O} = (\widetilde{y} - y)\,(\boldsymbol{W}_V \widetilde{x})^\top, \tag{31}$$

$$\frac{\partial \mathcal{L}}{\partial \boldsymbol{W}_V} = \boldsymbol{W}_O^\top\,(\widetilde{y} - y)\,\widetilde{x}^\top. \tag{32}$$

Under gradient flow, the dynamics are

$$\frac{d\boldsymbol{W}_{OV}}{dt} = (\widetilde{y} - y)\,\widetilde{x}^\top, \tag{33}$$

$$\frac{d\boldsymbol{W}_O}{dt} = (\widetilde{y} - y)\,\widetilde{x}^\top \boldsymbol{W}_V^\top, \tag{34}$$

$$\frac{d\boldsymbol{W}_V}{dt} = \boldsymbol{W}_O^\top\,(\widetilde{y} - y)\,\widetilde{x}^\top. \tag{35}$$

*Proof.* Equation (30) is the standard softmax cross-entropy gradient. Equations (31) (32) follow by chain rule, and (33)-(35) are the corresponding gradient-flow updates. □

### D.2. Lemma 1 Proof

**Lemma 1** [OV Parameters Balancedness]   Under gradient flow and balanced initialization,

$$\boldsymbol{W}_V(t)\boldsymbol{W}_V^\top(t) = \boldsymbol{W}_O^\top(t)\boldsymbol{W}_O(t) \qquad \forall t.$$

*Proof.* Using (34)–(35),

$$\begin{aligned}
\frac{d}{dt}\big(\boldsymbol{W}_V \boldsymbol{W}_V^\top\big) &= \boldsymbol{W}_V \left(\frac{d\boldsymbol{W}_V}{dt}\right)^\top + \left(\frac{d\boldsymbol{W}_V}{dt}\right)\boldsymbol{W}_V^\top \\
&= \boldsymbol{W}_V \widetilde{x}(\widetilde{y} - y)^\top \boldsymbol{W}_O + \boldsymbol{W}_O^\top(\widetilde{y} - y)\widetilde{x}^\top \boldsymbol{W}_V^\top.
\end{aligned} \tag{36}$$

Similarly,

$$\begin{aligned}
\frac{d}{dt}\big(\boldsymbol{W}_O^\top \boldsymbol{W}_O\big) &= \left(\frac{d\boldsymbol{W}_O}{dt}\right)^\top \boldsymbol{W}_O + \boldsymbol{W}_O^\top \left(\frac{d\boldsymbol{W}_O}{dt}\right) \\
&= \boldsymbol{W}_V \widetilde{x}(\widetilde{y} - y)^\top \boldsymbol{W}_O + \boldsymbol{W}_O^\top(\widetilde{y} - y)\widetilde{x}^\top \boldsymbol{W}_V^\top.
\end{aligned} \tag{37}$$

Comparing (36) and (37) gives

$$\frac{d}{dt}\big(\boldsymbol{W}_V \boldsymbol{W}_V^\top - \boldsymbol{W}_O^\top \boldsymbol{W}_O\big) = 0.$$

Hence the difference is constant in time, and under (approximately) balanced initialization it is 0, so $\boldsymbol{W}_V \boldsymbol{W}_V^\top = \boldsymbol{W}_O^\top \boldsymbol{W}_O$ for all $t$. □

### D.3. Lemma 2 Proof

**Lemma 2** [Relation Between Factorized and Collapsed OV parameterization]   Let $\widetilde{\boldsymbol{W}}_{OV} = \boldsymbol{W}_O \boldsymbol{W}_V$ and $\boldsymbol{W}_{OV}$ be the collapsed parameter. Then

$$\mathrm{vec}\!\left(\frac{\partial \widetilde{\boldsymbol{W}}_{OV}}{\partial t}\right) = \Omega(\boldsymbol{W}_{OV})\,\mathrm{vec}\!\left(\frac{\partial \boldsymbol{W}_{OV}}{\partial t}\right), \tag{38}$$

where

$$\Omega(C) := (C^\top C)^{1/2} \otimes I_M + I_d \otimes (CC^\top)^{1/2},$$

and $\Omega(\alpha C) = \alpha \, \Omega(C)$ (degree-1 homogeneous).

*Proof.* From Lemma 1, $\boldsymbol{W}_V \boldsymbol{W}_V^\top = \boldsymbol{W}_O^\top \boldsymbol{W}_O$. For fixed $t$, write $\boldsymbol{W}_O = \boldsymbol{U}_O \Sigma_O \boldsymbol{V}_O^\top$ and $\boldsymbol{W}_V = \boldsymbol{U}_V \Sigma_V \boldsymbol{V}_V^\top$. The Gram matrices share eigenvalues

$$\Sigma_O^\top \Sigma_O = \Sigma_V \Sigma_V^\top = diag(\rho_1 I_{d_1}, \ldots, \rho_m I_{d_m}),$$

hence $\Sigma_O = \Sigma_V = diag(\sqrt{\rho_1} I_{d_1}, \ldots, \sqrt{\rho_m} I_{d_m})$. Therefore,

$$\boldsymbol{W}_O \boldsymbol{W}_O^\top = \boldsymbol{U}_O diag(\rho_1 I_{d_1}, \ldots, \rho_m I_{d_m}) \boldsymbol{U}_O^\top, \tag{39}$$

$$\boldsymbol{W}_V^\top \boldsymbol{W}_V = \boldsymbol{V}_V diag(\rho_1 I_{d_1}, \ldots, \rho_m I_{d_m}) \boldsymbol{V}_V^\top. \tag{40}$$

Moreover, since $\boldsymbol{W}_{OV} = \boldsymbol{W}_O \boldsymbol{W}_V$,

$$\boldsymbol{W}_{OV} \boldsymbol{W}_{OV}^\top = \boldsymbol{U}_O diag(\rho_1^2 I_{d_1}, \ldots, \rho_m^2 I_{d_m}) \boldsymbol{U}_O^\top, \tag{41}$$

$$\boldsymbol{W}_{OV}^\top \boldsymbol{W}_{OV} = \boldsymbol{V}_V diag(\rho_1^2 I_{d_1}, \ldots, \rho_m^2 I_{d_m}) \boldsymbol{V}_V^\top. \tag{42}$$

Thus,

$$\boldsymbol{W}_O \boldsymbol{W}_O^\top = (\boldsymbol{W}_{OV} \boldsymbol{W}_{OV}^\top)^{1/2}, \qquad \boldsymbol{W}_V^\top \boldsymbol{W}_V = (\boldsymbol{W}_{OV}^\top \boldsymbol{W}_{OV})^{1/2}.$$

Using the product rule,

$$\frac{d\widetilde{\boldsymbol{W}}_{OV}}{dt} = \frac{d\boldsymbol{W}_O}{dt} \boldsymbol{W}_V + \boldsymbol{W}_O \frac{d\boldsymbol{W}_V}{dt}.$$

Substituting (34)-(35) and the identities above yields

$$\frac{d\widetilde{\boldsymbol{W}}_{OV}}{dt} = (y - \widetilde{y})\widetilde{\boldsymbol{x}}^\top (\boldsymbol{W}_{OV}^\top \boldsymbol{W}_{OV})^{1/2} + (\boldsymbol{W}_{OV} \boldsymbol{W}_{OV}^\top)^{1/2}(y - \widetilde{y})\widetilde{\boldsymbol{x}}^\top. \tag{43}$$

$$vec\left(\frac{d\widetilde{\boldsymbol{W}}_{OV}}{dt}\right) = \left((\boldsymbol{W}_{OV}^\top \boldsymbol{W}_{OV})^{1/2} \otimes I_M + I_d \otimes (\boldsymbol{W}_{OV} \boldsymbol{W}_{OV}^\top)^{1/2}\right) vec\left((y - \widetilde{y})\widetilde{\boldsymbol{x}}^\top\right). \tag{44}$$

Using (30) and (33) ,

$$vec\left(\frac{d\widetilde{\boldsymbol{W}}_{OV}}{dt}\right) = \Omega(\boldsymbol{W}_{OV}) \, vec\left(\frac{d\boldsymbol{W}_{OV}}{dt}\right),$$

which is (38). $\qquad\square$

## D.4. Proposition and Lemma for QK circuit

**Proposition 2.** *Consider* $\widehat{y} = \boldsymbol{W}_O \boldsymbol{W}_V \widetilde{x}$ *with* $\widetilde{x} = X^\top A$ *and* $A = \mathbb{S}(\mathbf{X} \boldsymbol{W}_K \boldsymbol{W}_Q^\top \boldsymbol{x}_T)$, *and define* $\boldsymbol{W}_{QK} = \boldsymbol{W}_K \boldsymbol{W}_Q^\top$.
*Let* $Z = Diag(A) - AA^\top$. *Then the gradient w.r.t.* $\boldsymbol{W}_{QK}$ *is*

$$\frac{\partial \mathcal{L}}{\partial \boldsymbol{W}_{QK}} = \boldsymbol{x}_T (\widetilde{y} - y)^\top \boldsymbol{W}_O \boldsymbol{W}_V \mathbf{X}^\top Z \mathbf{X} =: \boldsymbol{G}. \tag{45}$$

*By chain rule,*

$$\frac{\partial \mathcal{L}}{\partial \boldsymbol{W}_Q} = \boldsymbol{G}^\top \boldsymbol{W}_K, \tag{46}$$

$$\frac{\partial \mathcal{L}}{\partial \boldsymbol{W}_K} = \boldsymbol{G} \boldsymbol{W}_Q. \tag{47}$$

*Under gradient flow, the dynamics are*

$$\frac{d\boldsymbol{W}_{QK}}{dt} = \boldsymbol{G}, \tag{48}$$

$$\frac{d\boldsymbol{W}_Q}{dt} = \boldsymbol{G}^\top \boldsymbol{W}_K, \tag{49}$$

$$\frac{d\boldsymbol{W}_K}{dt} = \boldsymbol{G}\boldsymbol{W}_Q. \tag{50}$$

*Proof.* Equation (45) is the standard gradient with respect to the $\boldsymbol{W}_{QK}$ parameters. Equations (46) and (47) follow by chain rule, and (49) and (50) are the corresponding gradient-flow updates. $\qquad\square$

**Lemma 4** (QK Parameter Balancedness). *Let $\boldsymbol{W}_{QK} = \boldsymbol{W}_K \boldsymbol{W}_Q^\top$ and define $\boldsymbol{G}$ as in Equation (45). Under gradient flow and balanced initialization,*

$$\boldsymbol{W}_Q^\top(t)\boldsymbol{W}_Q(t) = \boldsymbol{W}_K^\top(t)\boldsymbol{W}_K(t) \qquad \forall t.$$

*Proof.*

$$\frac{d}{dt}\big(\boldsymbol{W}_Q^\top \boldsymbol{W}_Q\big) = \boldsymbol{W}_Q^\top \left(\frac{d\boldsymbol{W}_Q}{dt}\right) + \left(\frac{d\boldsymbol{W}_Q}{dt}\right)^\top \boldsymbol{W}_Q$$
$$= \boldsymbol{W}_Q^\top \boldsymbol{G}^\top \boldsymbol{W}_K + \boldsymbol{W}_K^\top \boldsymbol{G}\boldsymbol{W}_Q. \tag{51}$$

Similarly,

$$\frac{d}{dt}\big(\boldsymbol{W}_K^\top \boldsymbol{W}_K\big) = \boldsymbol{W}_K^\top \left(\frac{d\boldsymbol{W}_K}{dt}\right) + \left(\frac{d\boldsymbol{W}_Q}{dt}\right)^\top \boldsymbol{W}_Q$$
$$= \boldsymbol{W}_K^\top \boldsymbol{G}\boldsymbol{W}_Q + \boldsymbol{W}_Q^\top \boldsymbol{G}^\top \boldsymbol{W}_K. \tag{52}$$

$$\frac{d}{dt}\big(\boldsymbol{W}_Q^\top \boldsymbol{W}_Q\big) = \boldsymbol{W}_K^\top \boldsymbol{G}\boldsymbol{W}_Q + \boldsymbol{W}_Q^\top \boldsymbol{G}^\top \boldsymbol{W}_K, \tag{53}$$

$$\frac{d}{dt}\big(\boldsymbol{W}_K^\top \boldsymbol{W}_K\big) = \boldsymbol{W}_Q^\top \boldsymbol{G}^\top \boldsymbol{W}_K + \boldsymbol{W}_K^\top \boldsymbol{G}\boldsymbol{W}_Q. \tag{54}$$

Comparing (53) and (54) yields

$$\frac{d}{dt}\big(\boldsymbol{W}_Q^\top \boldsymbol{W}_Q - \boldsymbol{W}_K^\top \boldsymbol{W}_K\big) = 0.$$

Thus the difference is constant in time and equals 0 under balanced initialization, so $\boldsymbol{W}_Q^\top \boldsymbol{W}_Q = \boldsymbol{W}_K^\top \boldsymbol{W}_K$ for all $t$. $\qquad\square$

**Lemma 5** (QK preconditioning identity). *Let $\widetilde{\boldsymbol{W}}_{QK} = \boldsymbol{W}_K \boldsymbol{W}_Q^\top$ and $\boldsymbol{W}_{QK}$ be the collapsed parameter. Then*

$$\mathrm{vec}\left(\frac{\partial \widetilde{\boldsymbol{W}}_{QK}}{\partial t}\right) = \Omega(\boldsymbol{W}_{QK}) \, \mathrm{vec}\left(\frac{\partial \boldsymbol{W}_{QK}}{\partial t}\right), \tag{55}$$

*where*

$$\Omega(C) := (C^\top C)^{1/2} \otimes I_d + I_d \otimes (CC^\top)^{1/2},$$

*and $\Omega(\alpha C) = \alpha \, \Omega(C)$.*

*Proof.* Using Lemma 4, $\boldsymbol{W}_Q^\top \boldsymbol{W}_Q = \boldsymbol{W}_K^\top \boldsymbol{W}_K$, SVD argument of the proof of Lemma 2, we obtain

$$\boldsymbol{W}_Q \boldsymbol{W}_Q^\top = (\boldsymbol{W}_{QK}^\top \boldsymbol{W}_{QK})^{1/2}, \qquad \boldsymbol{W}_K \boldsymbol{W}_K^\top = (\boldsymbol{W}_{QK} \boldsymbol{W}_{QK}^\top)^{1/2}.$$

Using the product rule,

$$\frac{d\widetilde{\boldsymbol{W}}_{QK}}{dt} = \frac{d\boldsymbol{W}_K}{dt}\boldsymbol{W}_Q^\top + \boldsymbol{W}_K\left(\frac{d\boldsymbol{W}_Q}{dt}\right)^\top = \boldsymbol{G}\boldsymbol{W}_Q\boldsymbol{W}_Q^\top + \boldsymbol{W}_K\boldsymbol{W}_K^\top\boldsymbol{G}.$$

Thus,

$$\frac{d\widetilde{\boldsymbol{W}}_{QK}}{dt} = \boldsymbol{G}(\boldsymbol{W}_{QK}^\top\boldsymbol{W}_{QK})^{1/2} + (\boldsymbol{W}_{QK}\boldsymbol{W}_{QK}^\top)^{1/2}\boldsymbol{G}. \tag{56}$$

Vectorizing (56) and using $vec(AB) = (B^\top \otimes I)vec(A)$,

$$vec\left(\frac{d\widetilde{\boldsymbol{W}}_{QK}}{dt}\right) = \left((\boldsymbol{W}_{QK}^\top\boldsymbol{W}_{QK})^{1/2} \otimes I_d + I_d \otimes (\boldsymbol{W}_{QK}\boldsymbol{W}_{QK}^\top)^{1/2}\right)vec(\boldsymbol{G}).$$

Under gradient flow, (45) implies

$$vec\left(\frac{d\widetilde{\boldsymbol{W}}_{QK}}{dt}\right) = \Omega(\boldsymbol{W}_{QK})\, vec\left(\frac{d\boldsymbol{W}_{QK}}{dt}\right),$$

which is (55). $\qquad\square$

### D.5. Additional Results DTAP Heatmaps and Multilayer Setting Results

*Table 5.* Train and test performance together with attention-based interpretability metrics on real-world datasets. ↑ indicates higher is better, while ↓ indicates lower is better. All results are reported as mean over 5 random seeds. We report 95% confidence intervals over five random seeds in Table 6.

| Dataset | Setting | Train Performance ↑ | Test Performance ↑ | AC ↑ | ACMC ↑ | MRTA (Rollout) ↑ | Suff. ↓ | Compr. ↑ |
|---|---|---|---|---|---|---|---|---|
| HateXplain ( 2-Layers ) | Baseline | 94.58 | 58.12 | 4.54 | 0.00 | 0.08 | 0.49 | 0.22 |
| | Faster QK | 96.10 | 57.91 | 9.86 | 5.23 | 0.15 | 0.32 | 0.35 |
| HateXplain (4-Layers) | Baseline | 94.29 | 57.84 | 8.89 | 0.00 | 0.1 | 0.53 | 0.21 |
| | Faster QK | 95.89 | 58.51 | 12.89 | 6.45 | 0.16 | 0.44 | 0.26 |

*Table 6.* In this table we report 95% CI as ± values for each metric for Table 5.

| Dataset | Setting | Train Performance ↑ | Test Performance ↑ | AC ↑ | ACMC ↑ | MRTA (Rollout) ↑ | Suff. ↓ | Compr. ↑ |
|---|---|---|---|---|---|---|---|---|
| HateXplain ( 2-Layers ) | Baseline | 0.002 | 0.004 | 0.80 | 0.00 | 0.007 | 0.07 | 0.03 |
| | Faster QK | 0.006 | 0.02 | 1.20 | 1.39 | 0.005 | 0.1 | 0.05 |
| HateXplain (4-Layers) | Baseline | 0.008 | 0.009 | 1.45 | 0.00 | 0.001 | 0.06 | 0.01 |
| | Faster QK | 0.74 | 0.01 | 2.39 | 2.12 | 0.025 | 0.07 | 0.03 |

### D.6. Detailed Metric Discussion

We adopt the definitions of sufficiency and comprehensiveness from (DeYoung et al., 2020).

**Sufficiency**: Sufficiency captures whether the important tokens are sufficient to retain the original prediction.

$$\text{Sufficiency} = f(x) - f(r_{:k\%})$$

,

where $r : k\%$ denotes the top-$k\%$ most important tokens ranked by attention score.

**Comprehensiveness**: Comprehensiveness measures the change in the predicted-class probability after removing important tokens.

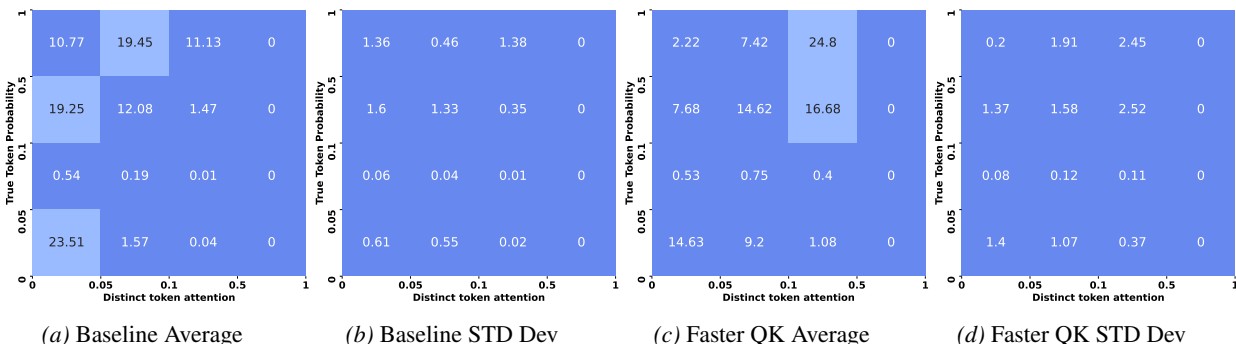

*Figure 5.* 6 layer model Distinct Token Attention-Prediction heat map of SQuAD QA Data (a)(b) Baseline Model Average and Standard Deviation (c)(d) FasterQK Model Average and Standard Deviation (5 runs)

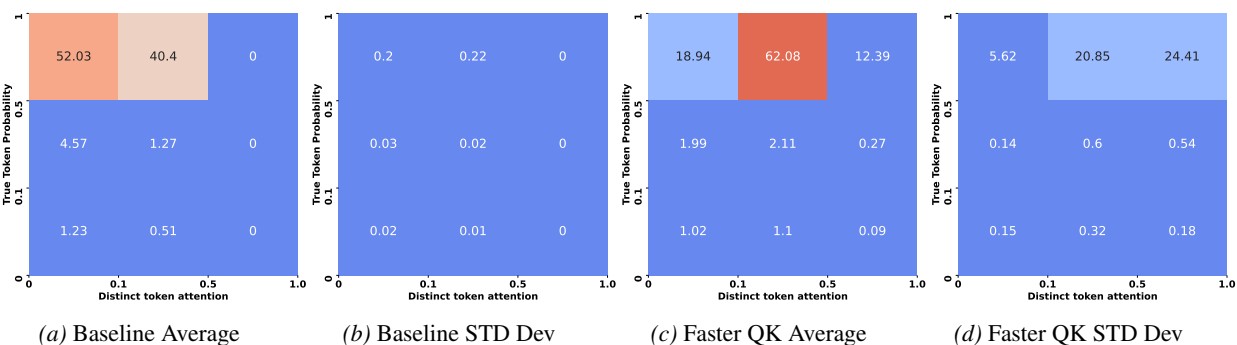

*Figure 6.* 1 layer model Distinct Token Attention-Prediction heat map of SVA data (a)(b) Baseline Model Average and Standard Deviation (c)(d) Faster QK Model Average and Standard Deviation (5 runs)

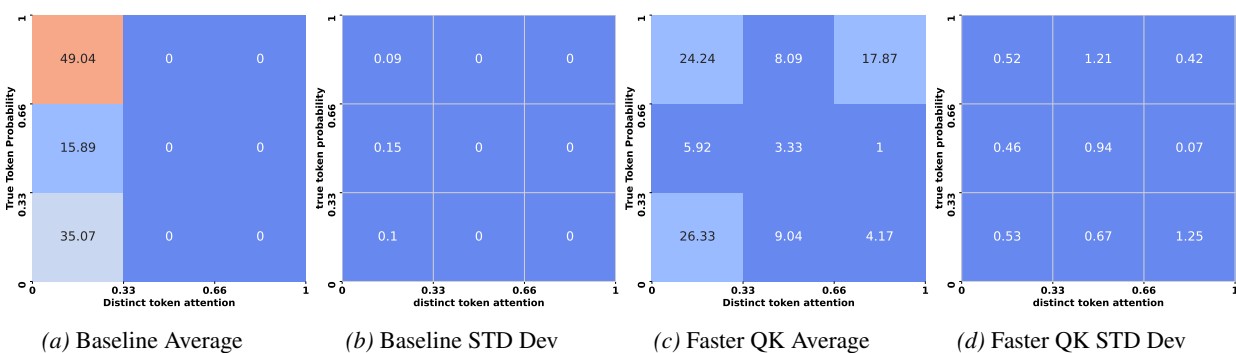

*Figure 7.* 1 layer model Distinct Token Attention-Prediction heat map of HateXplain data (a)(b) Baseline Model Average and Standard Deviation (c)(d) Faster QK Model Average and Standard Deviation (5 runs)

$$\text{Comprehensiveness} = f(x) - f(x/r_{:k\%})$$

where $r_{:k\%}$ refers to top-k% most important tokens chosen based on attention scores.

**Mean Relevant Token Attention (MRTA)**: Using the learned attention vector, we compute the average attention mass assigned to relevant tokens for an instance. This metric can be computed exactly using DTAP heatmaps, if the full attention distribution is known.

**Attention Confidence (AC)**: We compute fraction of instances where attention confidence is high. This refers to right-half column sum in DTAP heatmap. For example: Fig 5a for baseline model we will have AC value $11.13 + 1.47 + 0.01 + 0.04 =$ 12.65 (Table 1 column 5 row 1).

**Attention Confidence Model Confidence (ACMC)**: We compute fraction of instances where attention confidence is high as well as prediction probability is high. This refers to right-half columns and top-half rows sum in DTAP heatmap. For example: Fig 5a for baseline model we will have ACMC value $11.13 + 1.47 = 12.6$ (Table 1 column 5 row 1).

