# OpenReview forum: "Faster Query-Key Learning Sharpens Attention in Self-Attention Models"
_ICML.cc/2026/Conference — ICML 2026 regular_

### Official Review · Reviewer_m4rQ · 2026-02-20

**Soundness:** 3
**Presentation:** 2
**Significance:** 2
**Originality:** 2
**Overall Recommendation:** 4
**Confidence:** 3

**Summary:**

The paper investigates how different parameterizations (Factorized vs. Collapsed) of the Query-Key (QK) and Output-Value (OV) circuits in self-attention layers influence learning dynamics. Through gradient-flow analysis and empirical testing, the authors argue that the relative learning speed of the QK circuit is a primary driver of "attention sharpening." They propose that faster QK learning leads to more interpretable attention maps by concentrating mass on task-relevant tokens.

**Compliance With Llm Reviewing Policy:**

Affirmed.

**Final Justification:**

the authors clarified my concerns in the rebuttal, I recommend accept if the rest of the reviewers are positive about the work too.

**Key Questions For Authors:**

Please see the weakness section

**Limitations:**

yes

**Strengths And Weaknesses:**

**Strengths**:

- Methodological Rigor: The use of both average and standard deviation in the heatmaps provides a much clearer picture of the stability of the attention patterns than a single "cherry-picked" example.

- Novel Perspective: The distinction between factorized and collapsed parameterizations offers a fresh look at why standard Transformers (FAFO) might behave differently than theoretically "simpler" versions (CACO).

- Relevance: The work is timely, addressing the fundamental question of how optimization choices impact the interpretability of attention mechanisms.

**Weaknesses**:

- Gaps in Empirical Reporting: A concern is the lack of standard deviations in the result tables. Without these, it is difficult to assess the statistical significance of the performance differences between the various parameterizations.

- Weak Claims on Interpretability: While the paper claims that faster QK learning leads to "more interpretable" maps, this claim needs more grounded evidence. Simply being "sharper" does not always equate to being more correct or interpretable; human evaluations or more rigorous proxy tasks for interpretability would strengthen this argument. Please revise phrasing.

- Scope of Theoretical Analysis: The theoretical framework is strictly limited to single-layer models. While this is a standard starting point, the dynamics of multi-layer Transformers, where layers interact and error signals are backpropagated through multiple attention steps, could be significantly more complex and may even counteract the "sharpening" effect observed here. It is difficult to make a point about the "better" take on parameterization if only one layer effects are explored.

- Synthetic Data Reliance: Much of the mechanistic insight relies on highly structured synthetic data. It is unclear if these dynamics hold in "noisier" real-world environments where the distinction between "informative" and "common" tokens is fuzzy or context-dependent.

- Downstream Utility: The paper doesn't sufficiently demonstrate if "sharper" attention actually helps with downstream performance on difficult tasks (e.g., long-context reasoning). If sharpening doesn't improve performance, its value is purely academic/visual.

*Formatting & Clarity*:

- Table Formatting: Table 1, Table 4, and Table 5 all exceed the page margins. This is a violation of formatting rules and makes the paper difficult to read. Furthermore, table captions should be placed above the tables, not below.

- Figure Clarity: In Figure 2, the axes are too tightly packed and difficult to read. In Figure 7, the subfigures have inconsistent sizes, which detracts from the professional quality of the work.

- Mathematical Notation: On page 23, please check the punctuation following formulas (e.g., misplaced commas). The appendix should also follow a correct format.

---

> ### Author Rebuttal · Authors · 2026-03-31
>
> We thank the Reviewer m4rQ for the thoughtful and supportive review. We appreciate the reviewer's recognition of the paper's methodological rigor, novel perspective, and relevance.
>
> ### Weaknesses:
> **Gaps in Empirical Reporting:** The results in Table 1 are averaged over 5 random seeds, and the corresponding 95% confidence intervals are reported in Appendix Table 4. We will provide the 95% CI for Appendix Table 5 in the revised version. For both Tables 1 and 5, we find that the train and test performance of the baseline and the faster QK are comparable, with faster QK showing statistically significant improvements in interpretability metrics such as AC, ACMC, MRTA, Sufficiency, and Comprehensiveness.
>
> **Weak Claims on Interpretability:** We agree that attention sharpness does not imply more interpretability. We thus use multiple proxies of attention interpretability in the experiments. Tables 1 and 5 report Attention confidence (AC), Attention and Model Confidence (ACMC), and Mean Relevant Token Attention (MRTA), which are natural metrics that arise from DTAP heatmaps. In addition, we also report faithfulness metrics such as sufficiency and comprehensiveness (Deyoung et al. 2020). We will revise the phrasing to indicate these are merely proxies for attention interpretability.
>
>
> **Scope of Theoretical Analysis:** The theoretical analysis is restricted to the single-layer setting, where the dynamics of QK and OV circuit parameters can be characterized analytically. Empirically, however, we observe that the attention-sharpening effect persists beyond the single-layer setting. Table 1 presents SQuAD QA results with a 6-layer GPT model, and Table 5 reports multilayer results for the HateXplain dataset. We therefore view the single-layer theory as a tractable mechanistic account of the attention sharpening induced by faster QK learning, with supporting empirical evidence in multilayer settings.
>
>
> **Synthetic Data Reliance:** The synthetic setting allows us to isolate and study the interaction between QK and OV circuit parameters, enabling us to analyse the parameter dynamics without confounding effects. We observe attention-sharpening in real data in Tables 1 and 5. In SQuAD QA, the words in the answer span serve as the relevant tokens, whereas in HateXplain, token relevance is based on human-annotated rationales, which are inherently noisy and context-dependent. The real-data results suggest that the attention-sharpening trend observed in the synthetic setting persists even when token relevance is noisy.
>
> **Downstream Utility:** The paper's contribution is that attention structure can be systematically shaped through optimization dynamics, while predictive performance stays comparable. This is useful because it provides a way to influence attention structure in the model, enabling precise mechanistic analysis of which tokens the model relies on during prediction. Evaluating whether such control leads to better performance on difficult tasks such as long-context reasoning is an important direction for future work.
>
> **Formatting and Clarity:** We thank the reviewer for pointing out the formatting and clarity issues. We will correct the formatting of Tables and improve the clarity of the figures and appendix in the revised version. We will move the captions above all the tables in the revised version.

---

> > ### Author Rebuttal · Reviewer_m4rQ · 2026-04-01
> >
> > Thank you for the answers.

---

### Official Review · Reviewer_SFmV · 2026-03-03

**Soundness:** 3
**Presentation:** 3
**Significance:** 3
**Originality:** 3
**Overall Recommendation:** 4
**Confidence:** 1

**Summary:**

The goal of this paper is to explore the interpretability of attention during training. Through derivation and experiments, this paper explains the training characteristics of attention, providing new perspectives for researchers.

**Compliance With Llm Reviewing Policy:**

Affirmed.

**Final Justification:**

My concerns have been addressed. I decide to keep the rating.

**Key Questions For Authors:**

As shown in the 'Strengths and Weaknesses' section.

**Limitations:**

The author does not discuss the potential negative societal impacts. I believe this paper does not have potential negative societal impacts.

**Strengths And Weaknesses:**

I am not a researcher in this specific subfield, and the following are my non-professional opinions.

Strengths:
1. This paper's exploration of the research question is essential, with scientificity and universality. Therefore, the motivation of this paper is also persuasive;
2. The theoretical derivation in this paper is sufficient and can support its arguments;

Weaknesses:

Although this paper is theoretically oriented and its experiments are closely related to the theme, the number of experiments is slightly less compared to regular machine learning papers. This may affect readers' confidence and thus the dissemination of this paper in the community;

In addition, almost all large size tables are out of bounds. This is not a weakness, please consider correcting it.

---

> ### Author Rebuttal · Authors · 2026-03-31
>
> We thank the Reviewer SFmV for the thoughtful and supportive review.
>
>
> ### Weaknesses:
>
> **Weakness 1:** The paper’s goal is to study a mechanistic question: why models with similar predictive performance can exhibit very different learned attention patterns. We study how attention patterns are shaped by the relative optimization dynamics of QK and OV circuit parameters. From this perspective, the synthetic observations, mechanistic diagnosis, theoretical analysis in a tractable setting, and real-data validation are intended to provide a coherent chain of evidence for the proposed mechanism.
>
> Our real-data sharpness evaluation using AC, ACMC, and MRTA requires access to relevant token information. This limits the number of datasets that are directly suitable for such empirical evaluation. The idea of treating QK and OV circuit parameters differently during optimization has value beyond these datasets.
>
> **Table Formatting Issue:** We thank the reviewer for pointing out the table formatting issue. We will correct the issues in the revised version.

---

> > ### Author Rebuttal · Reviewer_SFmV · 2026-04-01
> >
> > Thanks for your rebuttal. It addresses my concerns and I will keep my positive score.

---

### Official Review · Reviewer_RK9J · 2026-03-07

**Soundness:** 4
**Presentation:** 2
**Significance:** 4
**Originality:** 3
**Overall Recommendation:** 4
**Confidence:** 5

**Summary:**

This paper studies a clear and worthwhile question: why can models with similar predictive performance exhibit very different learned attention patterns? Rather than revisiting the broad and often inconclusive debate over whether attention is interpretable, the paper focuses on a more specific mechanistic question about the optimization dynamics underlying attention structure. The central claim is that attention sharpness is not determined solely by predictive performance, but can be systematically shaped by the relative optimization dynamics of the QK and OV circuits. To support this claim, the paper presents a reasonably coherent chain of evidence: synthetic observations, diagnostic analysis with fixed attention, theoretical analysis in a tractable setting, and empirical validation on real data. Overall, this gives the paper more depth than a purely empirical study of attention patterns.

**Compliance With Llm Reviewing Policy:**

Affirmed.

**Final Justification:**

The authors have addressed my main concerns, especially by clarifying the intended scope of the empirical claims. This is an interesting and valuable work. I will maintain my overall recommendation with higher confidence.

**Key Questions For Authors:**

1. How robust is the main conclusion to initialization? Since the theory relies on near-zero and balanced initialization, could the authors provide a more systematic empirical study varying initialization scale and the degree of balancing? This would help clarify whether the proposed mechanism is primarily driven by relative learning speed, or whether it also depends materially on initialization-induced dynamics.
2. Can the authors provide more direct measurements of module-level learning dynamics? For example, it would strengthen the paper to track statistics such as parameter-update norms, function-space changes, or other module-level measures for QK and OV during training, rather than inferring relative speed mainly through learning-rate interventions.
3. Can the main empirical effect be reproduced through interventions other than learning-rate changes? For instance, it would be useful to know whether similar attention sharpening can also be induced by changing initialization scale, applying explicit gradient normalization, or using other module-level rescaling strategies. If qualitatively similar results hold across different interventions, that would more strongly support the claim that relative optimization speed is the key causal variable.

**Limitations:**

Partially. The paper would benefit from a clearer discussion that (1) sharper attention or improved alignment on annotation-based metrics should not be equated with fully trustworthy explanation, (2) the conclusions may be sensitive to the paper’s theoretical and experimental assumptions, and (3) optimization-induced changes in attention structure could make attention visualizations appear more interpretable without necessarily making them reliable for high-stakes decision support.

**Strengths And Weaknesses:**

Strengths
1. The research question is clear and well focused. The paper asks a concrete and important question: why do models with similar predictive performance often learn substantially different attention patterns? This is a strong framing because it shifts the discussion away from generic claims about attention interpretability toward a more precise question about optimization mechanisms.
2. The paper connects theory and experiments reasonably well. The work does not merely report an empirical phenomenon. Instead, it develops a fairly complete narrative: synthetic observations, mechanistic diagnosis, theoretical analysis in a simplified setting, and validation on real data. This gives the paper more explanatory depth than a purely descriptive study.
3. The decomposition of self-attention into QK and OV circuits is conceptually useful. Separating the query-key circuit from the output-value circuit helps distinguish between “where the model attends” and “how it uses the retrieved information.” This decomposition clarifies the optimization story and makes the analysis more interpretable.
4. The optimization-based perspective on attention structure is valuable. The paper’s main idea—that attention sharpness may be shaped by the relative optimization speeds of QK and OV, rather than by predictive performance alone—is interesting and potentially impactful. If correct, it suggests that attention patterns reflect not only task demands, but also systematic optimization biases.
5. The appendix proof chain is aligned with the main argument. From a content perspective, the appendix proofs are not just formulaic expansions. The main theorem reduces the training dynamics to a small number of analyzable scalar trajectories, and the subsequent lemmas study their growth behavior and the relationship between factorized and collapsed parameterizations.

Weakness:
1. The empirical notion of “relative optimization speed” is still largely proxy-based. In the theory section, the paper gives a reasonably clear gradient-flow treatment of the relative learning dynamics of QK and OV. However, in the empirical sections, the supporting evidence mainly comes from learning-rate interventions, matched-loss comparisons, and internal scalar/attention trajectories, rather than from direct measurements of submodule optimization progress, optimality gaps, or function-space progress. As a result, the experiments more strongly support an interpretation that is consistent with the relative-speed mechanism than a direct empirical confirmation of that quantity itself.
2. The intervention is suggestive, but it does not fully isolate the causal factor. The key intervention in the real-data experiments is to increase the learning rate for the QK parameter group. This is meaningful and does correspond to the theoretical ratio $r=\eta_{QK}/\eta_{OV}$. However, changing the learning rate can simultaneously affect parameter norm evolution, optimization noise, implicit regularization, and the timing of entering different nonlinear regimes. Therefore, the current evidence is better interpreted as showing that relative optimization speed is an important factor, rather than cleanly isolating it as the sole causal mechanism.
3. The theory is sensitive to initialization conditions, but empirical robustness to initialization is not fully explored. The theoretical connection between factorized and collapsed parameterizations relies on near-zero / balanced initialization, and the closed-form dynamics in the orthogonal setting are also derived under zero initialization of the effective parameters. In the main empirical comparison between the baseline and the faster-QK setting, the initial weights appear to be controlled, so the reported effect is not simply due to differing initializations. However, the paper does not systematically test robustness with respect to initialization scale, deviations from balancedness, or different initialization distributions. Initialization therefore seems to be an important part of the current mechanism, but not one that is fully characterized empirically.
4. The appendix proofs contain noticeable writing-quality issues that reduce the credibility and auditability of the theory section. Although the overall proof chain in the appendix does not appear fundamentally broken, its presentation contains anomalies that should not appear in a polished archival submission. For example, on page 21 (Lemma 3) and page 22 (Lemma A.5), the text contains the phrase “As in your derivation.” This does not read like formal exposition addressed to the reader, and instead, it sounds more like residue from conversational drafting, guided notes, intermediate derivation records, or even AI generated content. I do not want to over-interpret the authors’ writing process, but objectively, such phrasing weakens the professionalism and trustworthiness of the appendix as a finalized proof document. The authors should thoroughly clean up such residual language in a revision.
5. The numbering and cross-referencing of lemmas in the appendix are confusing. For instance, still around pages 20–22, the appendix uses labels such as Lemma 5, Lemma A.5, and Lemma 3 in a way that does not appear internally consistent. Even if this does not affect the mathematical conclusions directly, it makes the proof chain harder to follow and verify. For a theory-heavy paper, this kind of issue materially hurts readability and auditability. The appendix would benefit from a systematic pass over theorem/lemma/proposition numbering and cross-references.
6. There are several formatting and table-presentation issues, which lower the overall level of polish. Some tables appear to exceed the page width (e.g., Tables 1, 2, 4, and 5; this is not exhaustive, and the authors should check the full manuscript). The overfilled Table 1 in the main text might violate the ICML's policy because it should have captured more space. Though this does not affect my decision, the authors should check these carefully. Table captions should also be placed consistently above the table body. In addition, Table 4 looks visually rough, with border lines not fully closed, and the table styles are inconsistent across the paper: some tables use a plain grid style, while others are closer to a booktabs style. These issues do not change the main scientific claims, but they do reduce the professional quality of the submission and raise concerns about the level of care taken in preparing the appendix and experimental presentation. A full formatting pass would be beneficial.

---

> ### Author Rebuttal · Authors · 2026-03-31
>
> We thank Reviewer RK9J for the detailed and thoughtful review. We appreciate the reviewer's recognition that the paper addresses a clear and important research question, namely, why models with similar predictive performance can learn very different attention patterns. We also appreciate the reviewer's summary that our claims are supported by a coherent chain of evidence combining synthetic observations, theoretical analysis, and empirical validation.
>
> ### Weaknesses:
>
>
> **W1:** We agree that, in the real-data setting, relative optimization speed is not directly measured through a single module-level quantity. However, in the synthetic setting, we track the evolution of $\mu_{OV}(t)$, $\mu_{QK}(t)$, and the attention mass $\alpha(t)$ in Figure 4, both over time and at matched stopping losses. This provides a direct view of the QK/OV dynamics studied in theory. In the real-data experiments, our goal is not to directly measure any theoretical quantity, but to test whether an intervention on the ratio $r = \eta_{QK}/\eta_{OV}$ produces the predicted qualitative behavior.
>
>
> **W2:** Please see Q2 below.
>
> **W3:** Please see Q1 below.
>
> **W4, W5, and W6:** We thank the reviewer for pointing out the presentation issues in the appendix and tables. We will carefully revise the appendix, including removing residual language and correcting the lemmas' numbering. We will move the captions above all the tables and will correct the table formatting in the revised version.
>
>
> ### Key Questions for Authors:
>
> **Q1:** Yes, initialization is an important factor in the current theoretical analysis. In the reported baseline vs. increased-r comparisons, both settings use the same initialization scheme, so the observed sharpening effect is not due to different initial weights. Table 4 reports the 95% CIs for the metrics in Table 1 across 5 random initializations, showing that the results are stable.
> Initialization scale alone is unlikely to drive any systematic effect, but a systematic imbalance between the QK and OV circuit parameters may influence the optimization dynamics in a manner similar to the learning rate ratio. This remains an important direction for future work.
>
> **Q2:** We view the learning rate intervention as one concrete way to vary the relative optimization dynamics of the QK and OV circuits, rather than as the unique causal mechanism. In the synthetic setting, Theorem 1 characterizes the corresponding QK/OV evolution. Figure 4 tracks the evolution of theory-aligned scalar quantities $\mu_{QK}(t)$ and $\mu_{OV}(t)$, which parameterize the effective QK and OV trajectories over time. In the real-data experiments, the goal is to test whether intervening on these relative dynamics produces the predicted qualitative behavior. Other interventions that differentially affect the two circuits are interesting directions for future work (e.g., Weight decay, Initialization imbalance, etc.).
>
> **Q3:** In this work, we focus on the learning rate ratio $r=\eta_{QK}/\eta_{OV}$ because it is the most direct intervention on the relative optimization dynamics of QK and OV circuits. The theoretical framework suggests that other interventions that speed up QK optimization relative to OV may induce similar sharpening effects. We don’t claim this as a general result here. Alternative interventions, such as weight decay, initialization imbalance, or circuit-specific rescaling strategies, are natural directions for future work.
>
>
> **Limitations:** We will add a limitation section in the revised version, discussing the trustworthiness of sharper attention, the sensitivity of conclusions, and attention from an optimization perspective.

---

> > ### Author Rebuttal · Reviewer_RK9J · 2026-04-01
> >
> > Thank you for the responses. The authors have addressed my main concerns, especially by clarifying the intended scope of the empirical claims. This is an interesting and valuable work. I remain positive about the paper’s quality and contribution. I will maintain my overall recommendation with increased confidence.

---

### Official Review · Reviewer_7fJo · 2026-03-12

**Soundness:** 3
**Presentation:** 3
**Significance:** 2
**Originality:** 3
**Overall Recommendation:** 5
**Confidence:** 2

**Summary:**

This paper studies the relationship between query-key circuits and output-value circuits in attention heads. Namely, even if a set of attention heads exhibit the same external behavior (i.e., task accuracy), their inner mechanisms (i.e., attention patterns) can differ (ex: Figure 2). The authors study the source of such differences, and identify two big factors: one is the factorization vs. collapse of weight matrices (i.e., W_Q W_K^\top vs. a low rank W_{QK} weight matrix), while the other is the implicit relative learning rate for QK weights and OV weights.

In particular, the authors provide theoretical analyses using gradient-flow analysis to show that when the QK weights learn faster, the model compensates for the lag in OV by attending more heavily to the relevant tokens, thus resulting in sharper attention patterns. My understanding of the theories is that 1) the evolution of the effective matrix (W_{OV}) is not induced by arbitrarily re-scaling the factors W_O or W_V (Lemma 1) and that while each factor W_O and W_V follow gradient flow on L^2 (where L^2 is a loss function of W_O and W_V), their effective collapsed matrix W_{OV} follows the gradient flow of L^1 (where L^1 is a function of W_{OV}) with some pre-conditioning occurring first. Finally, Theorem 1/Lemma 3 states that under a set of assumptions (given a toy task with some constraints), the growth rate of the OV and QK circuits are either logarithmic or governed by the learning-rate ratio of QK/OV, respectively. Empirical experiments on this toy setting matches their theoretical predictions.

Lastly, the authors demonstrate an empirical study on a more realistic scenario of fine-tuning on 3 datasets, SQuAD, HateXplain, and subject-verb agreement (SVA), where they control for the learning rate ratio of QK and OV. The results match the author’s claims in that the end performance should not be affected much, whereas the attention patterns should look drastically different. Indeed, with a regime where QK is learned faster, the model ends up with very sharp attention patterns.

**Compliance With Llm Reviewing Policy:**

Affirmed.

**Final Justification:**

The rebuttal has addressed my concerns.

**Key Questions For Authors:**

I don’t understand lines 401-410: “In the increased r regime, we multiply the query-key learning rate by a fixed factor while keeping the output-value learning rate unchanged. We emphasize that adaptive optimization alone does not induce this behavior, as adaptive optimizers treat all parameter groups equally. We tune the query–key multiplier using validation performance to ensure stable training and comparable predictive accuracy across settings. Importantly, this tuning is not used to optimize attention metrics. “

Q: Does weight decay impact results?

Q: Minor formatting: Captions for tables should go above the tables (Table 1, 2, 3, 4).

**Limitations:**

The authors do not discuss limitations. For one, I think they could discuss the gap between their synthetic setting / assumptions versus realistic settings and discuss how one might close this gap / how their analyses might transfer in more realistic settings.

**Strengths And Weaknesses:**

Strengths:

The paper provides an abundance of theoretical analyses to study the learning dynamics of QK and OV circuits, with an emphasis on the relationship between the learning rate of the two. Their empirical evaluations match the predictions of the theory quite well, both in their synthetic setting (Figure 4) and more realistic setting (Table 1).

Weaknesses:

While the findings are fun, I think the paper could better motivate why we should care about qualitatively different attention patterns if, as the authors claim, the end behavior (i.e., task performance) ends up being the same. Understanding the learning dynamics and the relationship between QK and OV circuits is neat, but why is it important?

Second, there's a bit of a gap between their synthetic settings and more realistic settings. Namely, the task setup fits nicely into the analyses that the authors conduct, in that splitting up tokens into task-specific vs. generic sets, and further having class-dependent tokens necessarily entails that the model needs to learn to attend to such tokens, and then decode class-specific token information. I understand that this setting seems to have been studied before, but there’s a bit of a gap in this setting versus a more realistic setting of learning natural language (e.g., learning n-grams or hierarchically structured sequences, induction heads, etc.).
I also think Assumption 1 (task-specific token vectors are all mutually orthogonal) is quite strong of an assumption, especially given that often, token embeddings end up lying on a narrow cone. I think there’s a jump made between the toy task and real-world settings.

For the record, I don’t necessarily think this gap is a major weakness / reason for rejection, however.

---

> ### Author Rebuttal · Authors · 2026-03-30
>
> We thank Reviewer 7fJo for the detailed and thoughtful review. We appreciate the reviewer's recognition of our contributions, including the theoretical framework analyzing how circuit parameterization and learning-rate ratios shape learning dynamics.
>
> ### Weaknesses:
> **Weakness 1:** We agree with the reviewer that the motivation for qualitatively different attention patterns should be clearly discussed. Attention scores provide a view into the model's internal mechanism by indicating how information is selected and routed during prediction. From this perspective, our analysis is useful because it makes attention scores controllable and therefore useful for mechanistic analysis, such as which tokens a model relies on during prediction or how it behaves under perturbation. We anticipate that future work may use the idea to develop better algorithms that explicitly leverage internal attention for the final task.
>
> **Weakness 2:** We agree that the synthetic setting, Assumption 1, and experiments are chosen for analytical tractability and to measure the change in the intermediate outputs through attention. Our goal is to isolate the interaction between QK/OV learning dynamics in a setting where the mechanism can be studied cleanly. In section 7, we observe the same qualitative trend in more realistic settings that validate theory predictions. Extending our theoretical arguments to more relaxed assumptions (orthogonality of task-specific tokens) is an interesting area for future work.
>
>
> ### Key Questions For Authors:
> **Q1:** All our experiments use AdamW as the optimization algorithm. In the baseline, we use a single tuned base learning rate across parameter groups. AdamW intrinsically scales the gradient by a diagonal matrix, which is set by properties of the optimization landscape (estimated first- and second-order gradient moments). We explicitly have a higher QK learning rate via the parameter-group learning rate multiplier. This can be viewed as multiplying the diagonal preconditioning matrix of AdamW by another diagonal matrix with entries 1 or $r$ (the learning ratio), corresponding to the OV or QK circuit parameters. We will rephrase the sentence to clarify this.
>
> **Q2:** Having the same weight decay for QK and OV circuit parameters does not have any interesting behavior in our experiments (Fig 2 heatmaps remain similar). Having different weight decays for QK and OV circuit parameters might yield interesting results that we have not explored.
>
>
> **Q3:** We thank the reviewer for pointing out the formatting issue. We will move the captions above all the tables in the revised version.
>
> **Limitations:** We will include a limitations section discussing the gaps between synthetic and real-world settings.

---

> > ### Author Rebuttal · Reviewer_7fJo · 2026-03-31
> >
> > Thank you for your response and best of luck!

---

### Decision · Program_Chairs · 2026-04-30

**Decision:**

Accept (regular)

**Comment:**

The paper studies how the relative learning dynamics of query-key and output-value circuits shape attention patterns.  The authors use gradient-flow analysis to show that factorization induces an implicit rescaling of the two circuits' learning rates, and that faster QK learning leads to sharper attention at comparable predictive performance.  Paper is theoretically clean.

The theory is supported by synthetic experiments and fine-tuning results.  The main benefits from these experiments are improvements in the interpretability of the models.

Reviewers appreciated the theoretical framing. They raised concerns about
* the gap between the synthetic setting and realistic models,
*  the reliance on learning-rate interventions as a proxy for the underlying mechanism, and
* several presentation issues.

The authors clarified some of these concerns, including the scope of the empirical claims and the role of learning-rate interventions as one concrete way to probe the QK/OV dynamics.

I think it is still unclear whether sharper attention actually brings practical benefits. The paper shows that faster QK learning makes attention patterns sharper without hurting accuracy, but it does not show whether this sharpness translates into gains on downstream tasks.  The only measurable improvements are on attention-based interpretability metrics, which have known limitations as a proxy for model behavior (Jain & Wallace, 2019; Serrano & Smith, 2019).